# Interpretable Contrastive Monte Carlo Tree Search Reasoning

## Abstract

We propose (**S**)peculative (**C**)ontrastive **MCTS**\*: a novel Monte Carlo Tree Search (MCTS) reasoning algorithm for Large Language Models (LLMs) which significantly improves both reasoning accuracy and speed. Our motivation comes from: 1. Previous MCTS LLM reasoning works often overlooked its biggest drawback—slower speed compared to CoT; 2. Previous research mainly used MCTS as a tool for LLM reasoning on various tasks with limited quantitative analysis or ablation studies of its components from reasoning interpretability perspective. 3. The reward model is the most crucial component in MCTS, however previous work has rarely conducted in-depth study or improvement of MCTS's reward models. Thus, we conducted extensive ablation studies and quantitative analysis on components of MCTS, revealing the impact of each component on the MCTS reasoning performance of LLMs. Building on this, (i) we designed a highly interpretable reward model based on the principle of contrastive decoding and (ii) achieved an average speed improvement of 51.9% per node using speculative decoding. Additionally, (iii) we improved UCT node selection strategy and backpropagation used in previous works, resulting in significant performance improvement. We outperformed o1-mini by an average of 17.4% on the Blocksworld multi-step reasoning dataset using Llama-3.1-70B with SC-MCTS\*.

## 1 Introduction

With the remarkable development of Large Language Models (LLMs), models such as o1 (OpenAI, 2024a) have now gained a strong ability for multi-step reasoning across complex tasks and can solve problems that are more difficult than previous scientific, code, and mathematical problems. The reasoning task has long been considered challenging for LLMs. These tasks require converting a problem into a series of reasoning steps and then executing those steps to arrive at the correct answer. Recently, LLMs have shown great potential in addressing such problems. A key approach is using Chain of Thought (CoT) (Wei et al., 2024), where LLMs break down the solution into a series of reasoning steps before arriving at the final answer. Despite the impressive capabilities of CoT-based LLMs, they still face challenges when solving problems with an increasing number of reasoning steps due to the curse of autoregressive decoding (Sprague et al., 2024). Previous work has explored reasoning through the use of heuristic reasoning algorithms. For example, Yao et al. (2024) applied heuristic-based search, such as Depth-First Search (DFS) to derive better reasoning paths. Similarly, Hao et al. (2023) employed MCTS to iteratively enhance reasoning step by step toward the goal.

The tremendous success of AlphaGo (Silver et al., 2016) has demonstrated the effectiveness of the heuristic MCTS algorithm, showcasing its exceptional performance across various domains (Jumper et al., 2021; Silver et al., 2017). Building on this, MCTS has also made notable progress in the field of LLMs through multi-step heuristic reasoning. Previous work has highlighted the potential of heuristic MCTS to significantly enhance LLM reasoning capabilities. Despite these advancements, substantial challenges remain in fully realizing the benefits of heuristic MCTS in LLM reasoning.

Figure 1: An overview of SC-MCTS*. We employ a novel reward model based on the principle of contrastive decoding to guide MCTS Reasoning on Blocksworld multi-step reasoning dataset.

The first key challenge is that MCTS's general reasoning ability is almost entirely dependent on the reward model's performance (as demonstrated by our ablation experiments in Section 5.5), making it highly challenging to design dense, general yet efficient rewards to guide MCTS reasoning. Previous works either require two or more LLMs (Tian et al., 2024) or training epochs (Zhang et al., 2024a), escalating the VRAM and computational demand, or they rely on domain-specific tools (Xin et al., 2024a;b) or datasets (Qi et al., 2024), making it difficult to generalize to other tasks or datasets.

The second key challenge is that MCTS is significantly slower than Chain of Thoughts (CoT). CoT only requires designing a prompt of multi-turn chats (Wei et al., 2024). In contrast, MCTS builds a reasoning tree with 2–10 layers depending on the difficulty of the task, where each node in the tree represents a chat round with LLM which may need to be visited one or multiple times. Moreover, to obtain better performance, we typically perform 2–10 MCTS iterations, which greatly increases the number of nodes, leading to much higher computational costs and slower reasoning speed.

To address the these challenges, we went beyond prior works that treated MCTS as a tool and focused on analyzing and improving its components especially reward model. Using contrastive decoding, we redesigned reward model by integrating interpretable reward signals, clustering their prior distributions, and normalizing the rewards using our proposed prior statistical method. To prevent distribution shift, we also incorporated an online incremental update algorithm. We found that the commonly used Upper Confidence Bound on Trees (UCT) strategy often underperformed due to sensitivity to the exploration constant, so we refined it and improved backpropagation to favor steadily improving paths. To address speed issues, we integrated speculative decoding as a "free lunch." All experiments were conducted using the Blocksworld dataset detailed in Section 5.1.

Our goal is to: (i) design novel and high-performance reward models and maximize the performance of reward model combinations, (ii) analyze and optimize the performance of various MCTS components, (iii) enhance the interpretability of MCTS reasoning, (iv) and accelerate MCTS reasoning. Our contributions are summarized as follows:

1. We went beyond previous works who primarily treated MCTS as an tool rather than analyzing and improving its components. Specifically, we found the UCT strategy in most previous works may failed to function from our experiment. We also refined the backpropagation of MCTS to prefer more steadily improving paths, boosting performance.

2. To fully study the interpretability of MCTS multi-step reasoning, we conducted extensive quantitative analysis and ablation studies on every component. We carried out numerous experiments from both the numerical and distributional perspectives of the reward models, as well as its own interpretability, providing better interpretability for MCTS multi-step reasoning.

3. We designed a novel, general action-level reward model based on the principle of contrastive decoding, which requires no external tools, training, or datasets. Additionally, we found that previous works often failed to effectively harness multiple reward models, thus we proposed a statistical linear combination method. At the same time, we introduced speculative decoding to speed up MCTS reasoning by an average of 52% as a "free lunch."

We demonstrated the effectiveness of our approach by outperforming OpenAI's flagship o1-mini model by an average of 17.4% using Llama-3.1-70B on the Blocksworld multi-step reasoning dataset.

## 2 RELATED WORK

**Large Language Models Multi-Step Reasoning** One of the key focus areas for LLMs is understanding and enhancing their reasoning capabilities. Recent advancements in this area focused on developing methods that improve LLMs' ability to handle complex tasks in domains like code generation and mathematical problem-solving. Chain-of-Thought (CoT) (Wei et al., 2024) reasoning has been instrumental in helping LLMs break down intricate problems into a sequence of manageable steps, making them more adept at handling tasks that require logical reasoning. Building upon this, Tree-of-Thought (ToT) (Yao et al., 2024) reasoning extends CoT by allowing models to explore multiple reasoning paths concurrently, thereby enhancing their ability to evaluate different solutions simultaneously. Complementing these approaches, Monte Carlo Tree Search (MCTS) has emerged as a powerful reasoning method for decision-making in LLMs. Originally successful in AlphaGo's victory (Silver et al., 2016), MCTS has been adapted to guide model-based planning by balancing exploration and exploitation through tree-based search and random sampling, and later to large language model reasoning (Hao et al., 2023), showing great results. This adaptation has proven particularly effective in areas requiring strategic planning. Notable implementations like ReST-MCTS* (Zhang et al., 2024a), rStar (Qi et al., 2024), MCTSr (Zhang et al., 2024b) and Xie et al. (2024) have shown that integrating MCTS with reinforced self-training, self-play mutual reasoning or Direct Preference Optimization (Rafailov et al., 2023) can significantly improve reasoning capabilities in LLMs. Furthermore, recent advancements such as Deepseek Prover (Xin et al., 2024a;b) demonstrates the potential of these models to understand complex instructions such as formal mathematical proof.

**Decoding Strategies** Contrastive decoding and speculative decoding both require Smaller Language Models (SLMs), yet few have realized that these two clever decoding methods can be seamlessly combined without any additional cost. The only work that noticed this was Yuan et al. (2024a), but their proposed speculative contrastive decoding focused on token-level decoding. In contrast, we designed a new action-level contrastive decoding to guide MCTS reasoning, the distinction will be discussed further in Section 4.1. For more detailed related work please refer to Appendix B.

## 3 PRELIMINARIES

### 3.1 MULTI-STEP REASONING

A multi-step reasoning problem can be modeled as a Markov Decision Process (Bellman, 1957) $\mathcal{M} = (S, A, P, r, \gamma)$. $S$ is the state space containing all possible states, $A$ the action space, $P(s'|s, a)$ the state transition function, $r(s, a)$ the reward function, and $\gamma$ the discount factor. The goal is to learn *and* to use a policy $\pi$ to maximize the discounted cumulative reward $\mathbb{E}_{\tau \sim \pi} \left[ \sum_{t=0}^{T} \gamma^t r_t \right]$. For reasoning with LLMs, we are more focused on using an existing LLM to achieve the best reasoning.

### 3.2 MONTE CARLO TREE SEARCH

Monte Carlo Tree Search (MCTS) is a decision-making algorithm involving a search tree to simulate and evaluate actions. The algorithm operates in the following four phases:

**Node Selection:** The selection process begins at the root, selecting nodes hierarchically using strategies like UCT as the criterion to favor a child node based on its quality and novelty.

**Expansion:** New child nodes are added to the selected leaf node by sampling $d$ possible actions, predicting the next state. If the leaf node is fully explored or terminal, expansion is skipped.

**Simulation:** During simulation or "rollout", the algorithm plays out the "game" randomly from that node to a terminal state using a default policy.

**Backpropagation:** Once a terminal state is reached, the reward is propagated up the tree, and each node visited during the selection phase updates its value based on the simulation result.

Through iterative application of its four phases, MCTS efficiently improves reasoning through trials and heuristics, converging on the optimal solution.

### 3.3 CONTRASTIVE DECODING

We discuss vanilla Contrastive Decoding (CD) from Li et al. (2023), which improves text generation in LLMs by reducing errors like repetition and self-contradiction. CD uses the differences between an expert model and an amateur model, enhancing the expert's strengths and suppressing the amateur's weaknesses. Consider a prompt of length $n$, the CD objective is defined as:

$$\mathcal{L}_{\text{CD}}(x_{\text{cont}}, x_{\text{pre}}) = \log p_{\text{EXP}}(x_{\text{cont}}|x_{\text{pre}}) - \log p_{\text{AMA}}(x_{\text{cont}}|x_{\text{pre}})$$

where $x_{\text{pre}}$ is the sequence of tokens $x_1, \ldots, x_n$, the model generates continuations of length $m$, $x_{\text{cont}}$ is the sequence of tokens $x_{n+1}, \ldots, x_{n+m}$, and $p_{\text{EXP}}$ and $p_{\text{AMA}}$ are the expert and amateur probability distributions. To avoid penalizing correct behavior of the amateur or promoting implausible tokens, CD applies an adaptive plausibility constraint using an $\alpha$-mask, which filters tokens by their logits against a threshold, the filtered vocabulary $V_{\text{valid}}$ is defined as:

$$V_{\text{valid}} = \{i \mid s_{\text{EXP}}^{(i)} \geq \log \alpha + \max_k s_{\text{EXP}}^{(k)}\}$$

where $s_{\text{EXP}}^{(i)}$ and $s_{\text{AMA}}^{(i)}$ are unnormalized logits assigned to token i by the expert and amateur models. Final logits are adjusted with a coefficient $(1 + \beta)$, modifying the contrastive effect on output scores (Liu et al., 2021):

$$s_{\text{CD}}^{(i)} = (1 + \beta)s_{\text{EXP}}^{(i)} - s_{\text{AMA}}^{(i)}$$

However, our proposed CD is at action level, averaging over the whole action, instead of token level in vanilla CD. Our novel action-level CD reward more robustly captures the differences in confidence between the expert and amateur models in the generated answers compared to vanilla CD. The distinction will be illustrated in Section 4.1 and explained further in Appendix A.

### 3.4 SPECULATIVE DECODING AS "FREE LUNCH"

Based on Speculative Decoding (Leviathan et al., 2023), the process can be summarized as follows: Let $M_p$ be the target model with the conditional distribution $p(x_t|x_{<t})$, and $M_q$ be a smaller approximation model with $q(x_t|x_{<t})$. The key idea is to generate $\gamma$ tokens using $M_q$ and filter them against $M_p$'s distribution, accepting tokens consistent with $M_p$. Speculative decoding samples $\gamma$ tokens autoregressively from $M_q$, keeping those where $q(x) \leq p(x)$. If $q(x) > p(x)$, the sample is rejected with probability $1 - \frac{p(x)}{q(x)}$, and a new sample is drawn from the adjusted distribution:

$$p'(x) = \text{norm}(\max(0, p(x) - q(x))).$$

Since both contrastive and speculative decoding rely on the same smaller models, we can achieve the acceleration effect of speculative decoding as a "free lunch" (Yuan et al., 2024a).

## 4 METHOD

### 4.1 MULTI-REWARD DESIGN

Our primary goal is to design novel and and high-performance reward models for MCTS reasoning and to maximize the performance of reward model combinations, as our ablation experiments in Section 5.5 demonstrate that MCTS performance is almost entirely determined by the reward model.

SC-MCTS* is guided by three highly interpretable reward models: contrastive JS divergence, log-likelihood and self evaluation. Previous work such as (Hao et al., 2023) often directly adds reward functions with mismatched numerical magnitudes without any prior statistical analysis or linear combination. As a result, their combined reward models may fail to demonstrate full performance. Moreover, combining multiple rewards online presents numerous challenges such as distributional shifts in the values. Thus, we propose a statistically-informed reward combination method: **Multi-RM method**. Each reward model is normalized contextually by the fine-grained prior statistics of its empirical distribution. The pseudocode for reward model construction is shown in Algorithm 1. Please refer to Appendix D for a complete version of SC-MCTS* that includes other improvements such as dealing with distribution shift when combining reward functions online.

---

**Algorithm 1** SC-MCTS$^*$, reward model construction

---

**Input:** Expert LLM $\pi_e$, Amateur SLM $\pi_a$, Problem set $D$; $M$ selected problems for prior statistics, $N$ pre-generated solutions per problem, $K$ clusters

1:  $\tilde{A} \leftarrow$ Sample-solutions($\pi_e, D, M, N$)                    ▷ Pre-generate $M \times N$ solutions
2:  $p_e, p_a \leftarrow$ Evaluate($\pi_e, \pi_a, \tilde{A}$)                    ▷ Get policy distributions
3: **for** $r \in \{\text{JSD}, \text{LL}, \text{SE}\}$ **do**
4:      $\boldsymbol{\mu}_r, \boldsymbol{\sigma}_r, \boldsymbol{b}_r \leftarrow$ Cluster-stats($r(\tilde{A}), K$)          ▷ Prior statistics (Equation 1)
5:      $R_r \leftarrow x \mapsto (r(x) - \mu_r^{k^*})/\sigma_r^{k^*}$          ▷ Reward normalization (Equation 2)
6: **end for**
7: $R \leftarrow \sum_{r \in \{\text{JSD}, \text{LL}, \text{SE}\}} w_r R_r$                    ▷ Composite reward
8: $A_D \leftarrow$ MCTS-Reasoning($\pi_e, R, D, \pi_a$)                    ▷ Search solutions guided by $R$
**Output:** $A_D$

---

**Jensen-Shannon Divergence**    The Jensen-Shannon divergence (JSD) is a symmetric and bounded measure of similarity between two probability distributions $P$ and $Q$. It is defined as:

$$\text{JSD}(P \,\|\, Q) = \frac{1}{2}\text{KL}(P \,\|\, M) + \frac{1}{2}\text{KL}(Q \,\|\, M), \quad M = \frac{1}{2}(P + Q),$$

where $\text{KL}(P \,\|\, Q)$ is the Kullback-Leibler Divergence (KLD), and $M$ represents the midpoint distribution. The JSD is bounded between 0 and 1 for discrete distributions, making it better than KLD for online normalization of reward modeling.

Inspired by contrastive decoding, we propose our novel reward model: JSD between the expert model's logits and the amateur model's logits. Unlike vanilla token-level contrastive decoding (Li et al., 2023), our reward is computed at action-level, treating a sequence of action tokens as a whole:

$$R_{\text{JSD}} = \frac{1}{n} \sum_{i=T_{\text{prefix}}+1}^{n} [\text{JSD}(p_e(x_i|x_{<i}) \,\|\, p_a(x_i|x_{<i})]$$

where $n$ is the length of tokens, $T_{\text{prefix}}$ is the index of the last prefix token, $p_e$ and $p_a$ represent the softmax probabilities of the expert and amateur models, respectively. This approach ensures that the reward captures model behavior at the action level as the entire sequence of tokens is taken into account at once. This contrasts with vanilla token-level methods where each token is treated serially.

**Loglikelihood**    Inspired by Hao et al. (2023), we use a loglikelihood reward model to evaluate the quality of generated answers based on a given question prefix. The model computes logits for the full sequence (prefix + answer) and accumulates the log-probabilities over the answer part tokens.

Let the full sequence $x = (x_1, x_2, \ldots, x_{T_{\text{total}}})$ consist of a prefix and a generated answer. The loglikelihood reward $R_{\text{LL}}$ is calculated over the answer portion:

$$R_{\text{LL}} = \sum_{i=T_{\text{prefix}}+1}^{T_{\text{total}}} \log\left(\frac{\exp(z_\theta(x_i))}{\sum_{x' \in V} \exp(z_\theta(x'))}\right)$$

where $z_\theta(x_i)$ represents the unnormalized logit for token $x_i$. After calculating logits for the entire sequence, we discard the prefix and focus on the answer tokens to form the loglikelihood reward.

**Self Evaluation**    Large language models' token-level self evaluation can effectively quantify the model's uncertainty, thereby improving the quality of selective generation (Ren et al., 2023). We instruct the LLM to perform self evaluation on its answers, using a action level evaluation method, including a self evaluation prompt to explicitly indicate the model's uncertainty.

After generating the answer, we prompt the model to self-evaluate its response by asking "Is this answer correct/good?" This serves to capture the model's confidence in its own output leading to more informed decision-making. The self evaluation prompt's logits are then used to calculate a reward function. Similar to the loglikelihood reward model, we calculate the self evaluation reward $R_{\text{SE}}$ by summing the log-probabilities over the self-evaluation tokens.

**Harnessing Multiple Reward Models**  We collected prior distributions for the reward models and found some of them span multiple regions. Therefore, we compute the fine-grained prior statistics as mean and standard deviation of modes of the prior distribution $\mathcal{R} \in \{\mathcal{R}_{\text{JSD}}, \mathcal{R}_{\text{LL}}, \mathcal{R}_{\text{SE}}\}$:

$$\mu^{(k)} = \frac{1}{c_k} \sum_{R_i \in [b_1, b_{k+1})} R_i \quad \text{and} \quad \sigma^{(k)} = \sqrt{\frac{1}{c_k} \sum_{R_i \in [b_1, b_{k+1})} (R_i - \mu^{(k)})^2} \tag{1}$$

where $b_1 < b_2 < \cdots < b_{K+1}$ are the region boundaries in $\mathcal{R}$, $R_i \in \mathcal{R}$, and $c_k$ is the number of $R_i$ in $[b_1, b_{k+1})$. The region boundaries were defined during the prior statistical data collection phase 1.

After we computed the fine-grained prior statistics, the reward factors are normalized separately for each region (which degenerates to standard normalization if only a single region is found):

$$R_{\text{norm}}(x) = (R(x) - \mu^{(k^*)})/\sigma^{(k^*)}, \text{ where } k^* = \arg\max\{k : b_k \leq R(x)\} \tag{2}$$

This reward design, which we call **Multi-RM method**, has some caveats: first, to prevent distribution shift during reasoning, we update the mean and standard deviation of the reward functions online for each mode (see Appendix D for pseudocode); second, we focus only on cases with clearly distinct reward modes, leaving general cases for future work. For the correlation heatmap, see Appendix C.

## 4.2 Node Selection Strategy

Upper Confidence Bound applied on Trees Algorithm (UCT) (Coquelin & Munos, 2007) is crucial for the selection phase, balancing exploration and exploitation by choosing actions that maximize:

$$UCT_j = \bar{X}_j + C\sqrt{\frac{\ln N}{N_j}}$$

where $\bar{X}_j$ is the average reward of taking action $j$, $N$ is the number of times the parent has been visited, and $N_j$ is the number of times node $j$ has been visited for simulation, $C$ is a constant to balance exploitation and exploration.

However, $C$ is a crucial part of UCT. Previous work (Hao et al., 2023; Zhang et al., 2024b) had limited thoroughly investigating its components, leading to potential failures of the UCT strategy. This is because they often used the default value of 1 from the original proposed UCT (Coquelin & Munos, 2007) without conducting sufficient quantitative experiments to find the optimal $C$. This will be discussed in detail in Section 5.4.

## 4.3 Backpropagation

After each MCTS iteration, multiple paths from the root to terminal nodes are generated. By backpropagating along these paths, we update the value of each state-action pair. Previous MCTS approaches often use simple averaging during backpropagation, but this can overlook paths where the *goal achieved* metric $G(p)$ progresses smoothly (e.g., $G(p_1) = 0 \rightarrow 0.25 \rightarrow 0.5 \rightarrow 0.75$). These paths just few step away from the final goal $G(p) = 1$, are often more valuable than less stable ones.

To improve value propagation, we propose an algorithm that better captures value progression along a path. Given a path $\mathbf{P} = \{p_1, p_2, \ldots, p_n\}$ with $n$ nodes, where each $p_i$ represents the value at node $i$, the total value is calculated by summing the increments between consecutive nodes with a length penalty. The increment between nodes $p_i$ and $p_{i-1}$ is $\Delta_i = p_i - p_{i-1}$. Negative increments are clipped at $-0.1$ and downweighted by 0.5. The final path value $V_{\text{final}}$ is:

$$V_{\text{final}} = \sum_{i=2}^{n} \left\{ \begin{array}{ll} \Delta_i, & \text{if } \Delta_i \geq 0 \\ 0.5 \times \max(\Delta_i, -0.1), & \text{if } \Delta_i < 0 \end{array} \right\} - \lambda \times n \tag{3}$$

where $n$ is the number of nodes in the path and $\lambda = 0.1$ is the penalty factor to discourage long paths.

## 5 EXPERIMENTS

### 5.1 DATASET

Blocksworld (Valmeekam et al., 2024; 2023) is a classic domain in AI research for reasoning and planning, where the goal is to rearrange blocks into a specified configuration using actions like 'pick-up,' 'put-down,' 'stack,' and 'unstack. Blocks can be moved only if no block on top, and only one at a time. The reasoning process in Blocksworld is a MDP. At time step $t$, the LLM agent selects an action $a_t \sim p(a \mid s_t, c)$, where $s_t$ is the current block configuration, $c$ is the prompt template. The state transition $s_{t+1} = P(s_t, a_t)$ is deterministic and is computed by rules. This forms a trajectory of interleaved states and actions $(s_0, a_0, s_1, a_1, \ldots, s_T)$ towards the goal state.

One key feature of Blocksworld is its built-in verifier, which tracks progress toward the goal at each step. This makes Blocksworld ideal for studying heuristic LLM multi-step reasoning. However, we deliberately avoid using the verifier as part of the reward model as it is task-specific. More details of Blocksworld can be found in Appendix F.

### 5.2 MAIN RESULTS

To evaluate the SC-MCTS* algorithm in LLM multi-step reasoning, we implemented CoT, RAP-MCTS, and SC-MCTS* using Llama-3-70B and Llama-3.1-70B. For comparison, we used Llama-3.1-405B and GPT-4o for CoT, and applied 0 and 4 shot single turn for o1-mini, as OpenAI (2024b) suggests avoiding CoT prompting. The experiment was conducted on Blocksworld dataset across all steps and difficulties. For LLM settings, GPU and OpenAI API usage data, see Appendix E and H.

| Mode | Models | Method | Steps | | | | | | Avg. |
|---|---|---|---|---|---|---|---|---|---|
| | | | Step 2 | Step 4 | Step 6 | Step 8 | Step 10 | Step 12 | |
| Easy | Llama-3-70B ~Llama-3.2-1B | 4-shot CoT | 0.2973 | 0.4405 | 0.3882 | 0.2517 | 0.1696 | 0.1087 | 0.2929 |
| | | RAP-MCTS | 0.9459 | 0.9474 | 0.8138 | 0.4196 | 0.2136 | 0.1389 | 0.5778 |
| | | SC-MCTS* (Ours) | 0.9730 | 0.9737 | 0.8224 | 0.4336 | 0.2136 | 0.2222 | 0.5949 |
| | Llama-3.1-70B ~Llama-3.2-1B | 4-shot CoT | 0.5405 | 0.4868 | 0.4069 | 0.2238 | 0.2913 | 0.2174 | 0.3441 |
| | | RAP-MCTS | 1.0000 | 0.9605 | 0.8000 | 0.4336 | 0.2039 | 0.1111 | 0.5796 |
| | | SC-MCTS* (Ours) | 1.0000 | 0.9737 | 0.7724 | 0.4503 | 0.3010 | 0.1944 | 0.6026 |
| | Llama-3.1-405B | 0-shot CoT | 0.8108 | 0.6579 | 0.5931 | 0.5105 | 0.4272 | 0.3611 | 0.5482 |
| | | 4-shot CoT | 0.7838 | 0.8553 | 0.6483 | 0.4266 | 0.5049 | 0.4167 | 0.5852 |
| | o1-mini | 0-shot | 0.9730 | 0.7368 | 0.5103 | 0.3846 | 0.3883 | 0.1944 | 0.4463 |
| | | 4-shot | 0.9459 | 0.8026 | 0.6276 | 0.3497 | 0.3301 | 0.2222 | 0.5167 |
| | GPT-4o | 0-shot CoT | 0.5405 | 0.4868 | 0.3241 | 0.1818 | 0.1165 | 0.0556 | 0.2666 |
| | | 4-shot CoT | 0.5135 | 0.6579 | 0.6000 | 0.2797 | 0.3010 | 0.3611 | 0.4444 |
| Hard | Llama-3-70B ~Llama-3.2-1B | 4-shot CoT | 0.5556 | 0.4405 | 0.3882 | 0.2517 | 0.1696 | 0.1087 | 0.3102 |
| | | RAP-MCTS | 1.0000 | 0.8929 | 0.7368 | 0.4503 | 0.1696 | 0.1087 | 0.5491 |
| | | SC-MCTS* (Ours) | 0.9778 | 0.8929 | 0.7566 | 0.5298 | 0.2232 | 0.1304 | 0.5848 |
| | Llama-3.1-70B ~Llama-3.2-1B | 4-shot CoT | 0.6222 | 0.2857 | 0.3421 | 0.1722 | 0.1875 | 0.2174 | 0.2729 |
| | | RAP-MCTS | 0.9778 | 0.9048 | 0.7829 | 0.4702 | 0.1875 | 0.1087 | 0.5695 |
| | | SC-MCTS* (Ours) | 0.9778 | 0.9405 | 0.8092 | 0.4702 | 0.1696 | 0.2174 | 0.5864 |
| | Llama-3.1-405B | 0-shot CoT | 0.7838 | 0.6667 | 0.6053 | 0.3684 | 0.2679 | 0.2609 | 0.4761 |
| | | 4-shot CoT | 0.8889 | 0.6667 | 0.6579 | 0.4238 | 0.5804 | 0.5217 | 0.5915 |
| | o1-mini | 0-shot | 0.6889 | 0.4286 | 0.1776 | 0.0993 | 0.0982 | 0.0000 | 0.2034 |
| | | 4-shot | 0.9556 | 0.8452 | 0.5263 | 0.3907 | 0.2857 | 0.1739 | 0.4966 |
| | GPT-4o | 0-shot CoT | 0.6222 | 0.3929 | 0.3026 | 0.1523 | 0.0714 | 0.0000 | 0.2339 |
| | | 4-shot CoT | 0.6222 | 0.4167 | 0.5197 | 0.3642 | 0.3304 | 0.1739 | 0.4102 |

Table 1: Accuracy of various reasoning methods and models across steps and difficulty modes on the Blocksworld multi-step reasoning dataset.

From Table 1, it can be observed that SC-MCTS* significantly outperforms RAP-MCTS and 4-shot CoT across both easy and hard modes, and in easy mode, Llama-3.1-70B model using SC-MCTS* outperforms the 4-shot CoT Llama-3.1-405B model.

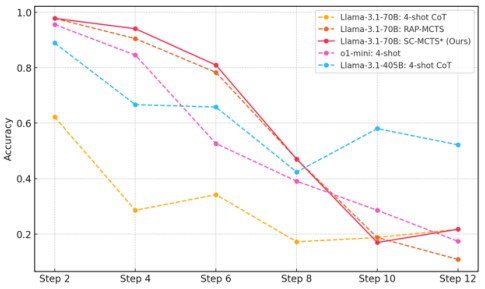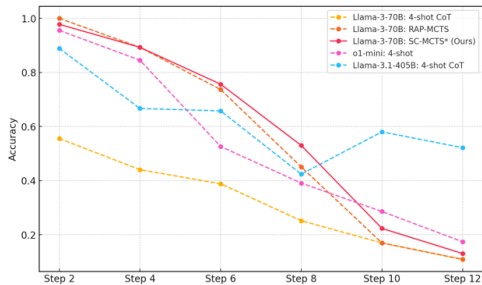

Figure 2: Accuracy comparison of various models and reasoning methods on the Blocksworld multi-step reasoning dataset across increasing reasoning steps.

From Figure 2, we observe that as the reasoning path lengthens, the performance advantage of two MCTS reasoning algorithms over themselves, GPT-4o, and Llama-3.1-405B's CoT explicit multi-turn chats and o1-mini implicit multi-turn chats (OpenAI, 2024b) in terms of accuracy diminishes, becoming particularly evident after Step 6. The accuracy decline for CoT is more gradual as the reasoning path extends, whereas models employing MCTS reasoning exhibits a steeper decline. This trend could be due to the fixed iteration limit of 10 across different reasoning path lengths, which might be unfair to longer paths. Future work could explore dynamically adjusting the iteration limit based on reasoning path length. It may also be attributed to our use of a custom EOS token to ensure output format stability in the MCTS reasoning process, which operates in completion mode. As the number of steps and prompt prefix lengths increases, the limitations of completion mode may become more pronounced compared to the chat mode used in multi-turn chats. Additionally, we observe that Llama-3.1-405B benefits significantly from its huge parameter size, although underperforming at fewer steps, experiences the slowest accuracy decline as the reasoning path grows longer.

## 5.3 REASONING SPEED

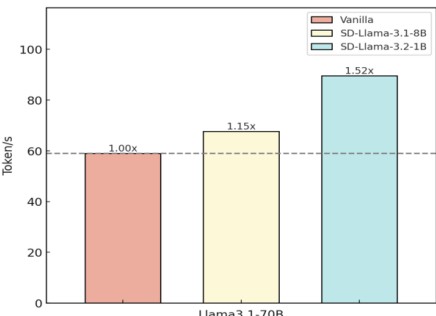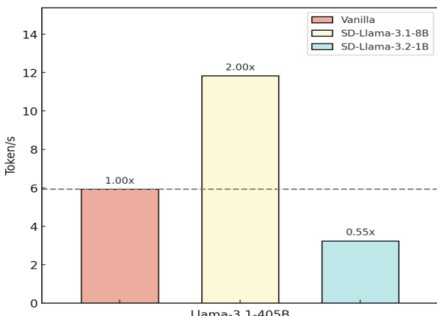

Figure 3: Speedup comparison of different model combinations. For speculative decoding, we use Llama-3.2-1B and Llama-3.1.8B as amateur models with Llama-3.1-70B and Llama-3.1-405B as expert models, based on average node-level reasoning speed in MCTS for Blocksworld multi-step reasoning dataset.

As shown in Figure 3, we can observe that the combination of Llama-3.1-405B with Llama-3.1-8B achieves the highest speedup, improving inference speed by approximately 100% compared to vanilla decoding. Similarly, pairing Llama-3.1-70B with Llama-3.2-1B results in a 51.9% increase in reasoning speed. These two combinations provide the most significant gains, demonstrating that speculative decoding with SLMs can substantially enhance node level reasoning speed. However, we can also observe from the combination of Llama-3.1-405B with Llama-3.2-1B that the parameters of SLMs in speculative decoding should not be too small, since the threshold for accepting draft tokens during the decoding process remains fixed to prevent speculative decoding from affecting performance (Leviathan et al., 2023), as overly small parameters may have a negative impact on decoding speed, which is consistent with the findings in Zhao et al. (2024); Chen et al. (2023).

## 5.4 PARAMETERS

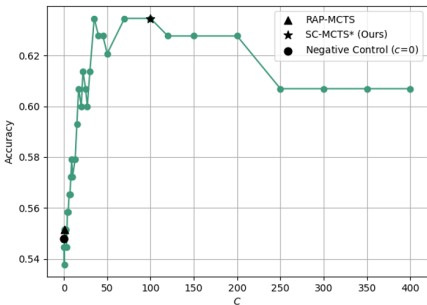
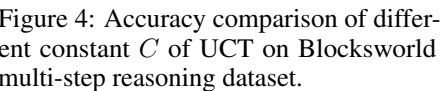

Figure 4: Accuracy comparison of different constant $C$ of UCT on Blocksworld multi-step reasoning dataset.

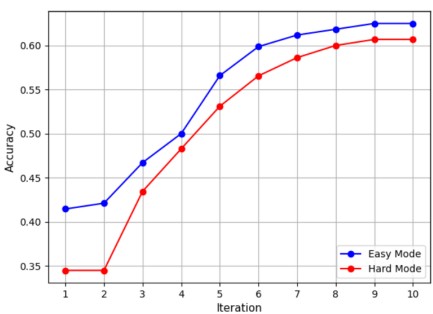

Figure 5: Accuracy comparison of different numbers of iteration on Blocksworld multi-step reasoning dataset.

As discussed in Section 4.2, the constant $C$ is a crucial part of UCT strategy, which completely determines whether the exploration term takes effect. Therefore, we conducted quantitative experiments on the constant $C$, to eliminate interference from other factors, we only use MCTS base with the common reward model $R_{\mathrm{LL}}$ for both RAP-MCTS and SC-MCTS$^*$. From Figure 4 we can observe that the constant $C$ of RAP-MCTS is too small to function effectively, while the constant $C$ of SC-MCTS$^*$ is the value most suited to the values of reward model derived from extensive experimental data. After introducing new datasets, this hyperparameter may need to be re-tuned.

From Figure 5, it can be observed that the accuracy of SC-MCTS$^*$ on multi-step reasoning increases steadily with the number of iterations. During the first 1-7 iterations, the accuracy rises consistently. After the 7th iteration, the improvement in accuracy becomes relatively smaller, indicating that under the experimental setting with depth limitations, the exponentially growing exploration nodes in later iterations bring diminishing returns in accuracy.

## 5.5 ABLATION STUDY

| Parts of SC-MCTS$^*$ | Accuracy (%) | Improvement (%) |
|---|---|---|
| MCTS base | 55.92 | — |
| + $R_{\mathrm{JSD}}$ | 62.50 | +6.58 |
| + $R_{\mathrm{LL}}$ | 67.76 | +5.26 |
| + $R_{\mathrm{SE}}$ | 70.39 | +2.63 |
| + Multi-RM Method | 73.68 | +3.29 |
| + Improved $C$ of UCT | 78.95 | +5.27 |
| + BP Refinement | 80.92 | +1.97 |
| **SC-MCTS$^*$** | 80.92 | Overall +25.00 |

Table 2: Ablation Study on the Blocksworld dataset at Step 6 under difficult mode. For a more thorough ablation study, the reward model for the MCTS base was set to pseudo-random numbers.

As shown in Table 2, the results of the ablation study demonstrate that each component of SC-MCTS$^*$ contributes significantly to performance improvements. Starting from a base MCTS accuracy of 55.92%, adding $R_{\mathrm{JSD}}$, $R_{\mathrm{LL}}$, and $R_{\mathrm{SE}}$ yields a combined improvement of 14.47%. Multi-RM method further boosts performance by 3.29%, while optimizing the $C$ parameter in UCT adds 5.27%, and the backpropagation refinement increases accuracy by 1.97%. Overall, SC-MCTS$^*$ achieves an accuracy of 80.92%, a 25% improvement over the base, demonstrating the effectiveness of these enhancements for complex reasoning tasks.

## 5.6 INTERPRETABILITY STUDY

In the Blocksworld multi-step reasoning dataset, we utilize a built-in ground truth verifier to measure the percentage of progress toward achieving the goal at a given step, denoted as $P$. The value of $P$ ranges between $[0, 1]$. For any arbitrary non-root node $N_i$, the progress is defined as:

$$P(N_i) = \text{Verifier}(N_i).$$

For instance, in a 10-step Blocksworld reasoning task, the initial node $A$ has $P(A) = 0$. After executing one correct action and transitioning to the next node $B$, the progress becomes $P(B) = 0.1$.

Given a non-root node $N_i$, transitioning to its parent node $\text{Parent}(N_i)$ through a specific action $a$, the contribution of $a$ toward the final goal state is defined as:

$$\Delta_a = P(\text{Parent}(N_i)) - P(N_i).$$

Next, by analyzing the relationship between $\Delta_a$ and the reward value $R_a$ assigned by the reward model for action $a$, we aim to reveal how our designed reward model provides highly interpretable reward signals for the selection of each node in MCTS. We also compare the performance of our reward model against a baseline reward model. Specifically, the alignment between $\Delta_a$ and $R_a$ demonstrates the interpretability of the reward model in guiding the reasoning process toward the goal state. Since Section 5.5 has already demonstrated that the reasoning performance of MCTS reasoning is almost entirely determined by the reward model, using interpretable reward models greatly enhances the interpretability of our algorithm SC-MCTS$^*$.

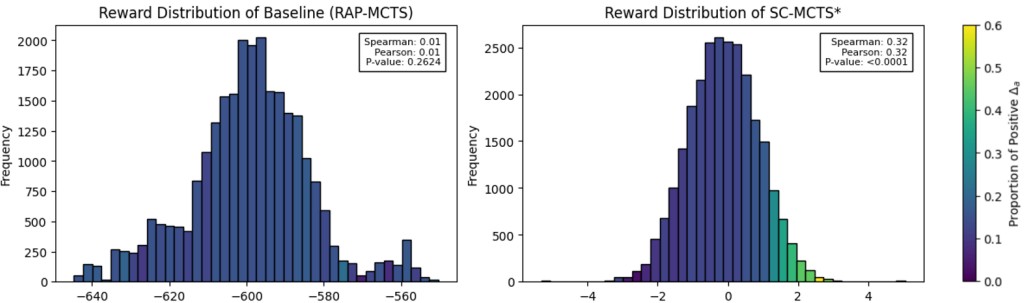

Figure 6: Reward distribution and interpretability analysis. The left histogram shows the baseline reward model (RAP-MCTS), while the right represents SC-MCTS$^*$. Bin colors indicate the proportion of positive $\Delta_a$ (lighter colors means higher proportions). Spearman and Pearson correlations along with p-values are shown in the top right of each histogram.

From Figure 6, shows that SC-MCTS* reward values correlate significantly with $\Delta_a$, as indicated by the high Spearman and Pearson coefficients. Additionally, the mapping between the reward value bins and the proportion of positive $\Delta_a$ (indicated by the color gradient from light to dark) is highly consistent and intuitive. This strong alignment suggests that our reward model effectively captures the progress toward the goal state, providing interpretable signals for action selection during reasoning.

These results highlight the exceptional interpretability of our designed reward model, which ensures that SC-MCTS* not only achieves superior reasoning performance but is also highly interpretable. This interpretability is crucial for understanding and improving the decision-making process in multi-step reasoning tasks, further validating transparency of our proposed algorithm.

## 6 CONCLUSION

In this paper, we present SC-MCTS$^*$, a novel and effective algorithm to enhancing the reasoning capabilities of LLMs. With extensive improvements in reward modeling, node selection strategy and backpropagation, SC-MCTS$^*$ boosts both accuracy and speed, outperforming OpenAI's o1-mini model by 17.4% on average using Llama-3.1-70B on the Blocksworld dataset. Experiments demonstrate its strong performance, making it a promising approach for multi-step reasoning tasks. For future work please refer to Appendix J. The synthesis of interpretability, efficiency and generalizability positions SC-MCTS$^*$ as a valuable contribution to advancing LLMs multi-step reasoning.

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

## A  ACTION-LEVEL CONTRASTIVE REWARD

We made the distinction between action-level variables and token-level variables: action-level (or step-level) variables are those that aggregate over all tokens in a reasoning step, and is typically utilized by the reasoning algorithm directly; token-level variables, by contrast, operates in a more microscopic and low-level environment, such as speculative decoding.

We found that the traditional contrastive decoding using the difference in logits, when aggregated over the sequence gives a unstable reward signal compared to JS divergence. We suspected this is due to the unbounded nature of logit difference, and the potential failure modes associated with it that needs extra care and more hyperparameter tuning.

## B   MORE RELATED WORK

**Large Language Models Multi-Step Reasoning**   Deepseek Prover (Xin et al., 2024a;b) relied on Lean4 as an external verification tool to provide dense reward signals in the RL stage. ReST-MCTS* (Zhang et al., 2024a) employed self-training to collect high-quality reasoning trajectories for iteratively improving the value model. AlphaLLM (Tian et al., 2024) used critic models initialized from the policy model as the MCTS reward model. rStar (Qi et al., 2024) utilized mutual consistency of SLMs and an additional math-specific action space. Xu (2023) proposed reconstructing fine-tuned LLMs into residual-based energy models to guide MCTS.

**Speculative Decoding**   Speculative decoding was first introduced in Leviathan et al. (2023), as a method to accelerate sampling from large autoregressive models by computing multiple tokens in parallel without retraining or changing the model structure. It enhances computational efficiency, especially in large-scale generation tasks, by recognizing that hard language-modeling tasks often include easier subtasks that can be approximated well by more efficient models. Similarly, DeepMind introduced speculative sampling (Chen et al., 2023), which expands on this idea by generating a short draft sequence using a faster draft model and then scoring this draft with a larger target model.

**Contrastive Decoding**   Contrastive decoding, as proposed by Li et al. (2023), is a simple, computationally light, and training-free method for text generation that can enhancethe quality and quantity by identifying strings that highlight potential differences between strong models and weak models. In this context, the weak models typically employ conventional greedy decoding techniques such as basic sampling methods, while the strong models are often well-trained large language models. This approach has demonstrated notable performance improvements in various inference tasks, including arithmetic reasoning and multiple-choice ranking tasks, thereby increasing the accuracy of language models. According to experiments conducted by O'Brien & Lewis (2023), applying contrastive decoding across various tasks has proven effective in enhancing the reasoning capabilities of LLMs.

## C   REWARD FUNCTIONS CORRELATION

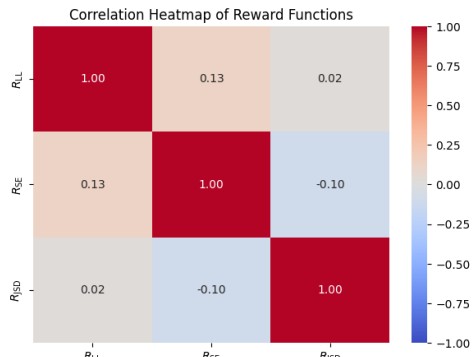

Figure 7: Reward Functions Correlation Heatmap.

It can be seen from Figure 7 that the correlations between the three reward functions are relatively low, absolute values all below 0.15. These low correlations of reward functions make them ideal for Multi-RM method.

## D   ALGORITHM DETAILS OF SC-MCTS*

The pseudocode inside MCTS reasoning of SC-MCTS* is shown in Algorithm 2, based on Zhang et al. (2024a). The complete version of SC-MCTS* is: first sample a subset of problems to obtain the prior data for reward values (Algorithm 1), then use it and two SLMs, one for providing contrastive reward signals, another for speculative decoding speedup, to perform MCTS reasoning. The changes of SC-MCTS* compared to previous works are highlighted in teal.

---

**Algorithm 2** SC-MCTS$^*$, reasoning

---

**Input:** expert LLM $\pi_e$, amatuer SLM $\pi_a$, speculative SLM $\pi_s$, problem $q$, reward model $R$, reward factor statistics $\mathcal{S}$, max iterations $T$, threshold $l$, branch $b$, rollout steps $m$, roll branch $d$, weight parameter $\alpha$, exploration constant $C$

1: $T_q \leftarrow$ Initialize-tree$(q)$
2: **for** $i = 1 \ldots T$ **do**
3:     $n \leftarrow$ Root$(T_q)$
4:     **while** $n$ is not leaf node **do**                 ▷ Node selection
5:         $n \leftarrow \arg\max_{n' \in \text{children}(n)}(v_{n'} + C\sqrt{\frac{\ln N_n}{N_{n'}}})$     ▷ Select child node based on UCT
6:     **end while**
7:     **if** $v_n \geq l$ **then break**                 ▷ Output solution
8:     **end if**
9:     **if** $n$ is not End of Inference **then**
10:         **for** $j = 1 \ldots b$ **do**                ▷ Thought expansion
11:            $n_j \leftarrow$ Get-new-child$(A_n, q, \pi_e)$     ▷ Expand based on previous steps
12:            $v_{n_j}, \mathcal{S} \leftarrow R(A_{n_j}, q, \pi_e, \pi_a, \mathcal{S})$    ▷ Evaluate contrastive reward and update reward factor statistics
13:         **end for**
14:         $n' \leftarrow \arg\max_{n' \in \text{children}(n)}(v_{n'})$
15:         $v_{\max} \leftarrow 0$
16:         **for** $k = 1 \ldots m$ **do**              ▷ Greedy MC rollout
17:            $A, v_{\max} \leftarrow$ Get-next-step-with-best-value$(A, q, \pi_e, \pi_s, d)$    ▷ Sample new children using speculative decoding and record the best observed value
18:         **end for**
19:         $v_{n'} \leftarrow \alpha v_{n'} + (1 - \alpha)v_{\max}$
20:         $N_{n'} \leftarrow N_{n'} + 1$          ▷ Update value and visit count of the rollout node
21:     **end if**
22:     Back-propagate$(n)$             ▷ Update value of parent nodes (Equation 3)
23: **end for**
24: $n \leftarrow$ Get-best-node$(T_q)$        ▷ Fetch the node with the highest value in the search tree

**Output:** $A_n$

---

Although we sampled a small portion of the dataset as prior data for reward values, distribution shift may still occur when normalizing reward values during reasoning. Therefore, we use the following algorithm to incrementally update the mean and standard deviation of the online reward distribution:

---

**Algorithm 3** Online incremental update of reward factor statistics

---

**Input:** reward factors $\mathcal{R}(= \{\text{JSD}, \text{LL}, \text{SE}\})$, statistics $\{\mu_r^{(k)}, \sigma_r^{(k)}, n_r^{(k)}\}_{r \in \mathcal{R}, k \in \{1, \ldots, K\}}$, cluster assignment function $f$

1: **for** $r \in \mathcal{R}$ **do**
2:     $k^* \leftarrow f(x)$               ▷ Assign sample to cluster
3:     $v_r \leftarrow r(x)$              ▷ Compute reward factor value
4:     $n_r^{(k^*)} \leftarrow n_r^{(k^*)} + 1$         ▷ Update sample count
5:     $\delta \leftarrow v_r - \mu_r^{(k^*)}$         ▷ Compute difference from mean
6:     $\mu_r^{(k^*)} \leftarrow \mu_r^{(k^*)} + \delta/n_r^{(k^*)}$      ▷ Update mean
7:     $M_2 \leftarrow (n_r^{(k^*)} - 1)(\sigma_r^{(k^*)})^2 + \delta(v_r - \mu_r^{(k^*)})$
8:     $\sigma_r^{(k^*)} \leftarrow \sqrt{M_2/n_r^{(k^*)}}$      ▷ Update standard deviation
9: **end for**

**Output:** updated statistics $\{\mu_r^{(k)}, \sigma_r^{(k)}, n_r^{(k)}\}_{r \in \mathcal{R}, k \in \{1, \ldots, K\}}$

---

# E  EXPERIMENTAL SETTINGS

For reproducibility, you can download the checkpoints from the Huggingface repository below and use the hyperparameters below. We utilized 4-bit quantized checkpoints in all experiments, as they only result in around 2% performance loss while providing several-fold reductions in memory usage and significantly improving inference speed (Frantar et al., 2022). For better output formatting to capture a single step and convert it into an MCTS node, we used the LLM's completion mode so we set LLM to greedy sampling, and we don't have to set an additional system prompt, simply apply prompts in Appendix F. Our experiments were all conducted on exllamav2 inference framework.

## E.1  CHECKPOINTS

| Usage | Models | Links |
|---|---|---|
| **Expert** | **Llama-3.1-405B** | `https://huggingface.co/hugging-quants/Meta-Llama-3.1-405B-Instruct-GPTQ-INT4` |
| | **Llama-3.1-70B** | `https://huggingface.co/hugging-quants/Meta-Llama-3.1-70B-Instruct-GPTQ-INT4` |
| | **Llama-3-70B** | `https://huggingface.co/TechxGenus/Meta-Llama-3-70B-Instruct-GPTQ` |
| **Amateur** | **Llama-3.1-8B** | `https://huggingface.co/hugging-quants/Meta-Llama-3.1-8B-Instruct-GPTQ-INT4` |
| | **Llama-3-8B** | `https://huggingface.co/astronomer/Llama-3-8B-Instruct-GPTQ-4-Bit` |
| | **Llama-3.2-1B** | `https://huggingface.co/meta-llama/Llama-3.2-1B` |
| **OpenAI** | **GPT-4o** | `https://platform.openai.com/docs/models/gpt-4o` |
| | **o1-mini** | `https://platform.openai.com/docs/models/o1` |

Table 3: Checkpoints used in experiments and their links.

## E.2  HYPERPARAMETERS

| Hyperparameter | Value |
|---|---|
| temperature | 1.0 |
| top-k | 1.0 |
| top-p | 1.0 |
| repetition_penalty | 1.0 |
| max_new_tokens | 200 |
| max_seq_len | 32768 |
| MCTS EOS: `Llama-3 family` | `"\n["` |
| CoT EOS: `Llama-3 family` | `"\n", "<|eot_id|>"` |

Table 4: LLM Hyperparameters and EOS tokens used in experiments.

# F  BLOCKSWORLD DATASET

The Blocksworld dataset comprises 600 instances with varying block numbers and plan lengths. Simpler instances have 3-5 blocks, while more complex cases involve up to 25 blocks, introducing additional goals and obstacles. This setup covers a range of problem difficulties for evaluating planning algorithms.

## F.1  DIFFICULTY SETTINGS

According to settings of LLM Reasoners (Hao et al., 2024), we divide the original 600 instances of Blocksworld (Valmeekam et al., 2024) into two parts, Easy and Hard settings.

In the Easy Blocksworld setting, we use more friendly demonstration cases. If a problem requires a specific minimum number of steps to solve, we select other problems that require the same number of steps as demonstration cases in the context. For example, if a problem requires at least 4 steps to solve, we use other 4-step problems as demonstration examples. For each group of problems, we randomly select 10 cases to create a pool of demonstration cases, while the remaining cases form the test set (a total of 540 cases). During inference, we randomly sample 4-shot demonstration cases from this pool to construct the prompts.

In the Hard Blocksworld setting, we randomly select 10 cases from the entire dataset to create the demonstration pool. These selected cases are then excluded from the test set, leaving a total of 590 cases for testing. During inference, we randomly sample 4-shot demonstration cases from this global pool, without considering the minimum number of actions required for the test case. For example, if a problem requires at least 4 steps to solve, we may still use demonstration cases that require a different number of steps, such as 2 or 12, as there is no restriction based on the number of actions.

---

**domain_intro:**
**I am playing with a set of objects. Here are the actions I can do:**
pick up a block
unstack a block from on top of another block
put down a block
stack a block on top of another block

**I have the following restrictions on my actions:**
To perform the Pick Up action, the block must be clear, on the table, and my hand must be empty. Once the Pick Up action is performed, I am holding the block, and my hand is no longer empty.

To perform the Unstack action, the block must be clear, on top of another block, and my hand must be empty. Once the Unstack action is performed, I am holding the block, and my hand is no longer empty.

To perform the Put Down action, I must be holding a block. Once the Put Down action is performed, the block is on the table, my hand is empty, and the block becomes clear.

To perform the Stack action, I must be holding a block, and the block I want to stack it on must be clear. Once the Stack action is performed, the block is on top of another block, my hand is empty, and the block on top is no longer clear.

---

Table 5: Normal Blocksworld Task Setting

## F.2 PROMPTS SETTINGS OF EASY BLOCKSWORLD

---

**Input Instructions:**
I am playing with a set of blocks where I need to arrange the blocks into stacks. Here are the actions I can do:

1. Pick up a block

2. Unstack a block from on top of another block

3. Put down a block

4. Stack a block on top of another block

I have the following restrictions on my actions:

1. I can only pick up or unstack one block at a time.

2. I can only pick up or unstack a block if my hand is empty.

3. I can only pick up a block if the block is on the table and the block is clear. A block is clear if the block has no other blocks on top of it and if the block is not picked up.

4. I can only unstack a block from on top of another block if the block I am unstacking was really on top of the other block.

5. I can only unstack a block from on top of another block if the block I am unstacking is clear.

Once I pick up or unstack a block, I am holding the block.

1. I can only put down a block that I am holding.

2. I can only stack a block on top of another block if I am holding the block being stacked.

3. I can only stack a block on top of another block if the block onto which I am stacking the block is clear.

Once I put down or stack a block, my hand becomes empty.

[STATEMENT]
As initial conditions I have that, the red block is clear, the hand is empty, the blue block is on top of the orange block, the red block is on the table, the orange block is on the table and the yellow block is on the table.
My goal is to have that the orange block is on top of the blue block. My plan is as follows:
[End Of STATEMENT]

[PLAN]
unstack the blue block from on top of the orange block
put down the blue block
pick up the orange block
stack the orange block on top of the blue block
[PLAN END]

[STATEMENT]
As initial conditions I have that, the red block is clear, the yellow block is clear, the hand is empty, the red block is on top of the blue block, the yellow block is on top of the orange block, the blue block is on the table and the orange block is on the table.
My goal is to have that the orange block is on top of the red block. My plan is as follows:
[End Of STATEMENT]

**Output format:**
[PLAN]
**[LLM Completion]**
[PLAN_END]

---

Table 6: The Prompt Settings for Easy Blocksworld

972
973
974
975
976

F.3   PROMPTS SETTINGS OF HARD BLOCKSWORLD

977
978
979
980

**Input Instructions:**
I am playing with a set of blocks where I need to arrange the blocks into stacks. Here are the actions I can do:

981     1. Pick up a block

982     2. Unstack a block from on top of another block

983     3. Put down a block

984     4. Stack a block on top of another block

985
986
I have the following restrictions on my actions:

987     1. I can only pick up or unstack one block at a time.

988     2. I can only pick up or unstack a block if my hand is empty.

989     3. I can only pick up a block if the block is on the table and the block is clear. A block
990        is clear if the block has no other blocks on top of it and if the block is not picked
991        up.

992     4. I can only unstack a block from on top of another block if the block I am unstacking
993        was really on top of the other block.

994     5. I can only unstack a block from on top of another block if the block I am unstacking
995        is clear.

996
Once I pick up or unstack a block, I am holding the block.

997     1. I can only put down a block that I am holding.

998     2. I can only stack a block on top of another block if I am holding the block being
999        stacked.

1000    3. I can only stack a block on top of another block if the block onto which I am
1001       stacking the block is clear.

1002
1003
Once I put down or stack a block, my hand becomes empty.

1004    [STATEMENT]
1005    As initial conditions I have that, the blue block is clear, the hand is empty, the blue block is
1006    on top of the red block, the red block is on the table, the orange block is on the table and the
1007    yellow block is on the table.
1008    My goal is to have that the blue block is on top of the orange block. My plan is as follows:
1009    [End Of STATEMENT]

1010    [PLAN]
1011    unstack the blue block from on top of the red block
1012    stack the blue block on top of the orange block
1013    [PLAN END]

1014    [STATEMENT]
1015    As initial conditions I have that, the red block is clear, the yellow block is clear, the hand
1016    is empty, the red block is on top of the blue block, the yellow block is on top of the orange
1017    block, the blue block is on the table and the orange block is on the table.
1018    My goal is to have that the orange block is on top of the red block. My plan is as follows:
1019    [End Of STATEMENT]

1020    **Output format:**
1021    [PLAN]
1022    **[LLM Completion]**
1023    [PLAN_END]

1024
1025

Table 7: The Prompt Settings for Hard Blocksworld

## G  EXAMPLE TREES OF DIFFERENT $c$ OF UCT

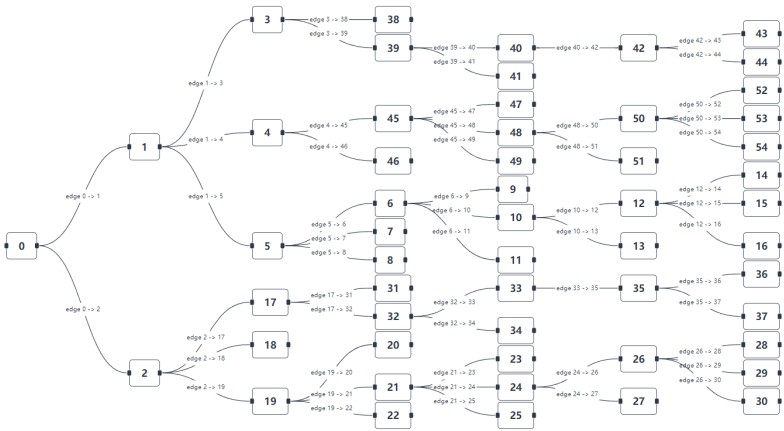

Figure 8: Monte Carlo Tree with origin parameter $c$ of UCT

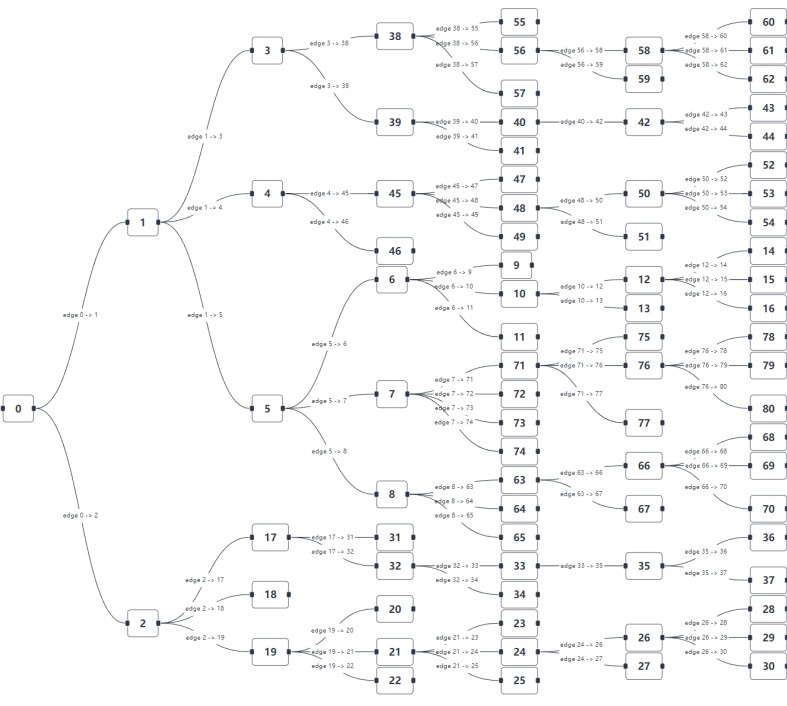

Figure 9: Monte Carlo Tree with our optimized parameter $c$ of UCT

From Figure 8 and 9 we can observed that with our optimized parameter $c$ of UCT, MCTS algorithm in node selection decisions tends to prioritize exploring new nodes rather than repeatedly following old paths, which may often lead to dead ends.

## H  OPENAI API DATA

| Difficulty | Model | USD per instance | Total Experiment Cost (USD) |
|---|---|---|---|
| **Easy (0-shot)** | GPT-4o | $0.0032 | $1.73 |
| | o1-mini | $0.0136 | $7.34 |
| **Easy (4-shot)** | GPT-4o | $0.0062 | $3.35 |
| | o1-mini | $0.0171 | $9.23 |
| **Hard (0-shot)** | GPT-4o | $0.0032 | $1.89 |
| | o1-mini | $0.0177 | $10.44 |
| **Hard (4-shot)** | GPT-4o | $0.0063 | $3.70 |
| | o1-mini | $0.0172 | $10.15 |

Table 8: OpenAI API cost of experiments on the Blocksworld dataset.

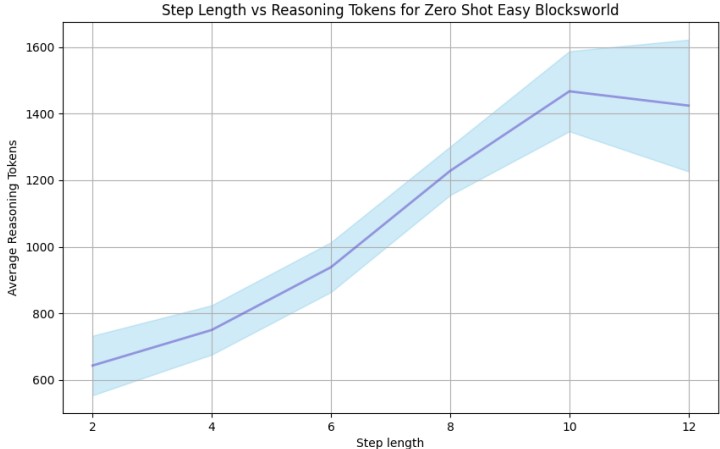

Figure 10: o1-mini Step Length vs Reasoning Tokens for Zero Shot in Easy Blocksworld

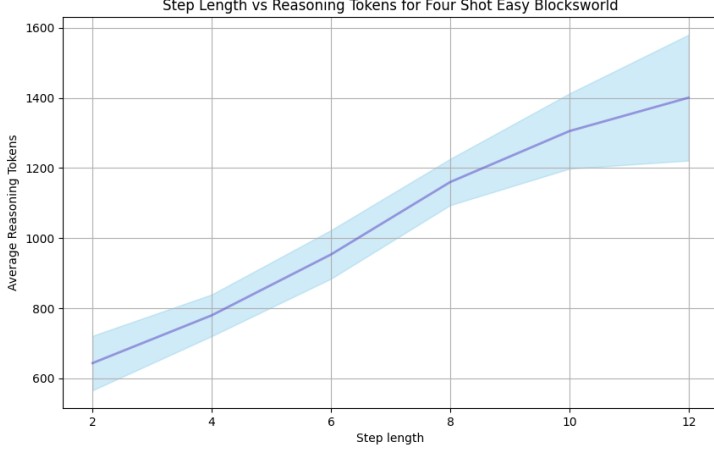

Figure 11: o1-mini Step Length vs Reasoning Tokens for Four Shot in Easy Blocksworld

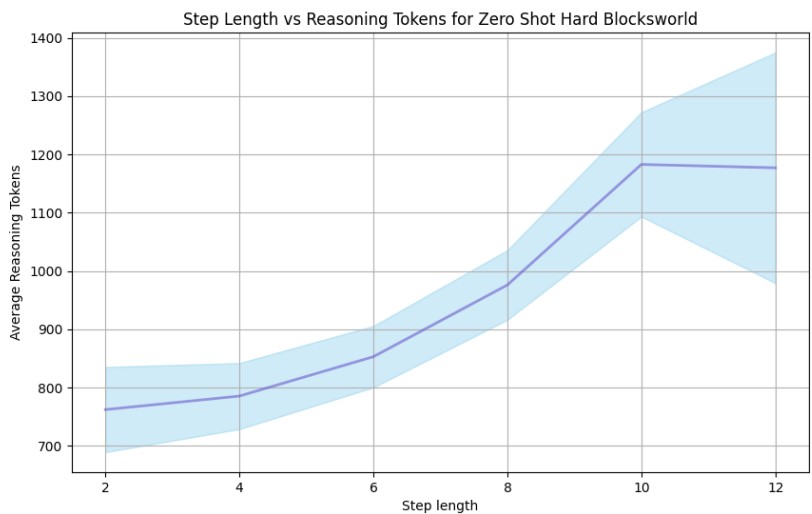

Figure 12: o1-mini Step Length vs Reasoning Tokens for Zero Shot in Hard Blocksworld

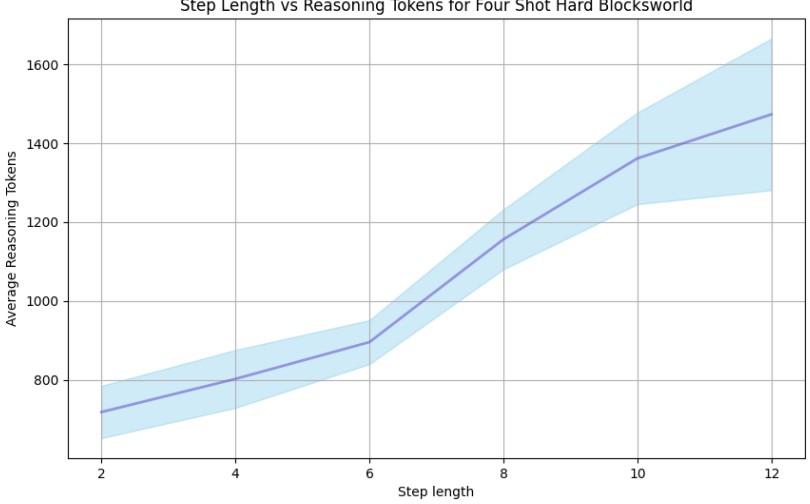

Figure 13: o1-mini Step Length vs Reasoning Tokens for Four Shot in Hard Blocksworld

## I  GPU USAGE

In the main experiments, the total GPU usage (measured in GPU hours) for different models on NVIDIA H800 SXM5 80GB GPUs shows a clear progression with model size. For RAP-MCTS, Llama-3 70B requires approximately 420 GPU hours across all steps and difficulty modes, Llama-3.1 70B model requires approximately 450 GPU hours. For SC-MCTS*, Llama-3 70B requires approximately 280 GPU hours across all steps and difficulty modes and difficulty modes, Llama-3.1 70B model requires approximately 300 GPU hours. For CoT, Llama-3-70B and Llama-3.1-70B both takes approximately 7 GPU hours across all steps and difficulty modes, while Llama-3.1 405B model exhibits significantly higher GPU usage, amounting to approximately 75 GPU hours. In the parameter research and algorithm development phase before main experiments, we consumed a total of around 800 GPU hours on NVIDIA A100 SXM4 80GB GPUs.

## J  FUTURE WORK

In future work, we can explore utilizing more metrics-based reward models (such as the three reward models discussed in this paper) with LM-based reward models (such as Critic LLM (McAleese et al., 2024) and Eurus (Yuan et al., 2024b)). Additionally, there is potential to design more general methods for splitting steps in other tasks and datasets. Since step-splitting is the most challenging part of MCTS multi-step reasoning generalization, although we conducted extensive experiments on the Blocksworld multi-step reasoning dataset, which is the most suitable dataset for studying MCTS multi-step reasoning as far as we know. Some previous works have attempted to use datasets like GSM8K and MATH through extensive adaptation efforts on the datasets themselves, however, we aim to design a more general method from the perspective of step-splitting. We hope that MCTS multi-step reasoning will achieve the same level of generalization as CoT, which remains a fundamental area for future research. Future work can also attempt to combine this approach with the fine-grained compositional reasoning framework (Chen et al., 2024) to further explore the boundaries of MCTS multi-step reasoning capabilities.

