# OpenReview forum: "Interpretable Contrastive Monte Carlo Tree Search Reasoning"
_ICLR.cc/2025/Conference — ICLR 2025 Conference Withdrawn Submission_

### Official Review · Reviewer_LCn7 · 2024-11-02

**Soundness:** 3
**Presentation:** 3
**Contribution:** 4
**Rating:** 8
**Confidence:** 3

**Summary:**

The paper introduces SC-MCTS∗, a novel enhancement to the Monte Carlo Tree Search algorithm, designed to improve the reasoning capabilities of Large Language Models (LLMs). It combines Contrastive Decoding and Speculative Decoding to not only increase reasoning accuracy but also to significantly reduce the time consumption.

**Strengths:**

see question

**Weaknesses:**

see question

**Questions:**

This paper presents a comprehensive Monte Carlo Tree Search (MCTS) method, SC-MCTS∗, which not only enhances the reasoning accuracy of Large Language Models (LLMs) but also reduces the time required for reasoning. It offers a novel and effective algorithm that significantly boosts both the accuracy and speed of reasoning, as demonstrated through extensive experiments and quantitative analysis. The authors have designed their experiments to include various models and reasoning methods, providing a robust comparison and validation of their proposed SC-MCTS∗ method.

However, a minor shortcoming of the experiments is the absence of performance evaluation on common mathematical test sets, such as GSM8k and MATH. Incorporating these benchmarks would have provided a more comprehensive assessment of the algorithm's generalization capabilities and its effectiveness across different types of reasoning tasks.

---

### Official Review · Reviewer_CGgt · 2024-11-03

**Soundness:** 2
**Presentation:** 2
**Contribution:** 2
**Rating:** 3
**Confidence:** 4

**Summary:**

This paper introduces SC-MCTS* (Speculative Contrastive MCTS), a novel Monte Carlo Tree Search algorithm for LLMs that addresses previous speed and reasoning accuracy limitations. The authors enhance MCTS through three key innovations: a contrastive decoding-based reward model, speculative decoding for 51.9% faster node processing, and improved UCT node selection and backpropagation strategies. Using Llama-3.1-70B, their method achieved a 17.4% performance improvement over o1-mini on the Blocksworld multi-step reasoning dataset.

**Strengths:**

- Shows significant improvement compared to the baseline
- Accelerates the search process by using speculative decoding

**Weaknesses:**

- The article is unclear, especially in section 4 where the authors fail to describe how CONTRASTIVE DECODING and SPECULATIVE DECODING are applied in MCTS.
- Using MCTS in LLM reasoning is not a novel approach, as numerous papers have already discussed its application to LLMs. This paper doesn't present any innovative ideas to enhance reasoning capabilities. Although the authors claim speculative decoding as one of their contributions, this inference acceleration paradigm was already mentioned in [1].
- The experimental evaluation is limited to only Blocksworld tasks, without exploring more complex reasoning problems such as MATH and HumanEval (code generation).
- The comparison with baselines is limited, lacking comparisons with other search-based methods like TOT, beam search[2], and Q*[3].

[1]:AlphaZero-Like Tree-Search can Guide Large Language Model Decoding and Training

[2]:Scaling LLM Test-Time Compute Optimally can be More Effective than Scaling Model Parameters

[3]:Q*: Improving Multi-step Reasoning for LLMs with Deliberative Planning

**Questions:**

- Could you further describe the MCTS process, especially how it integrates with speculative decoding?
- Compared to different search algorithms, particularly heuristic algorithms based on A*, what are the advantages of MCTS?
- The authors need to compare the efficiency and effectiveness (also known as test time scaling law) of different inference time methods to demonstrate how SC-MCTS* better balances the trade-off
- Why does using two models (Expert and Amateur) yield better results in the search context? Perhaps need to further explain where the improvements lie when combining Contrastive Decoding with MCTS compared to using only Expert model for MCTS

---

> ### Author Response · Authors · 2024-11-21
>
> **Response(1/3)**
>
> We would like to express our sincere gratitude for your careful reading and insightful feedback on our paper.
>
> **For weakness 1:**
>
> Thanks for your feedback! We have improved the explanation of the contrastive decoding and speculative decoding sections. Please see lines 230-248 and lines 197-198 in our revised version.
>
> **For weakness 2:**
>
> Thank you for your suggestions. As you mentioned, our major contribution is not about applying speculative decoding on MCTS, as noted in our paper, prior work has primarily used MCTS as a tool in some downstream reasoning tasks, with several technical reports published. However, quantitative studies and ablation analyses of MCTS components remain limited. Our contributions are as follows:
>
> 1. As shown in the ablation study in Section 5.5, the performance of MCTS reasoning is almost entirely dependent on the reward model. Thus, our key contribution is designing a **novel and high-performance reward model** based on the idea of contrastive decoding.
>
> 2. We identified flaws in previous algorithms that combined multiple reward models and proposed a **linear combination algorithm based on prior statistical methods**, supplemented by an **online incremental update algorithm for mean and variance** to prevent distributional shifts.
>
> 3. We observed that the UCT strategy may fail in prior work, potentially leading MCTS into dead ends. We improved this aspect.
>
> 4. We optimized the MCTS backpropagation algorithm to favor **steadily improving paths**, significantly enhancing performance.
>
> 5. In Section 5.6, we demonstrated that our reward model is **highly interpretable** compared to prior work by analyzing numerical distributions, quantile-reward mappings, Spearman correlation, Pearson correlation, and p-values. Consequently, **our SC-MCTS\*** is also highly interpretable since its performance is almost entirely determined by the reward model.
>
> All these contributions are emphasized in the **Introduction Section**. We have updated the title of the speculative decoding section to "Speculative Decoding as **'Free Lunch'**." As the title suggests, the acceleration benefits of speculative decoding are a "free lunch" provided by our novel reward model, which is designed based on the idea of contrastive decoding.
>
> **For weakness 3, 4:**
>
> In the Future Work section, we have addressed why our experiments were conducted exclusively on the Blocksworld multi-step reasoning dataset. This is because other reasoning datasets you mentioned, such as GSM8K and MATH, lack a unified implementation for task step segmentation and cannot operate MCTS reasoning using completion mode. All our experiments on Blocksworld were executed in completion mode, where each action can be easily segmented using a custom EOS token (e.g., for Llama-3, it’s “\n[”). This allows us to construct the search tree with ease, making the MCTS experiments highly controllable and enabling a more focused study of our reward model at the action level. These features make Blocksworld an ideal dataset for investigating LLM MCTS reasoning. Additionally, it includes a built-in ground truth verifier.
>
> We have not found other datasets with these properties that are also suitable for algorithms like Q\* (although there does not have an open-source implementation), making adaptation challenging. And if we were to adapt these datasets such as GSM8K, defining the action space for mathematical reasoning tasks would be very complex. We might need to rewrite the entire dataset to provide a unified step-segmentation template or fine-tune the model to enable natural and controllable step segmentation. This is left as future work.

---

> ### Author Response · Authors · 2024-11-21
>
> **Response(2/3)**
>
> **For question 1:**
>
> MCTS is a decision search algorithm based on stochastic simulations, commonly used for game decisions or problems with large search spaces. It is suitable for scenarios such as games, decision planning, reasoning, and other tasks requiring simulations of complex dynamic systems. MCTS does not require explicit goal states; instead, it evaluates simulations to derive optimal strategies. We have added practical MCTS examples in Appendix G for better understanding. In our Blocksworld setup, the objective is to guide the LLM in reasoning about stacking blocks to achieve a target state, given an initial block configuration and the goal configuration. There are exactly four available actions and only one correct reasoning path. Each state corresponds to a node in the tree, and by applying an action, the process moves to the next node. Through several iterations, the state progressively approaches the goal state. The algorithm operates in the following four phases:
>
> **Node Selection:** The selection process begins at the root, selecting nodes hierarchically using strategies like UCT as the criterion to favor a child node based on its quality and novelty.
>
> **Expansion:** New child nodes are added to the selected leaf node by sampling $d$ possible actions and predicting the next state. If the leaf node is fully explored or terminal, expansion is skipped.
>
> **Simulation:** During simulation or “rollout,” the algorithm plays out the “game” randomly from that node to a terminal state using a default policy.
>
> **Backpropagation:** Once a terminal state is reached, the reward is propagated up the tree, and each node visited during the selection phase updates its value based on the simulation result.
>
> Through iterative application of its four phases, MCTS efficiently improves reasoning through trials and heuristics, converging on the optimal solution. Typically, 10 iterations are required to achieve ideal performance. The first three iterations often behave as nearly random Monte Carlo processes. For each node, we calculate the associated reward value. After each iteration, the rewards are backpropagated to the root, leaving information about which paths hold higher value (cumulative rewards) for subsequent iterations to explore. However, we observed that previous MCTS approaches often use simple averaging during backpropagation, which can overlook paths where the **goal achieved** metric $G(p)$ progresses smoothly (e.g., $G(p_1) = 0 \rightarrow 0.25 \rightarrow 0.5 \rightarrow 0.75$). These paths, being only a few steps away from the final goal $G(p) = 1$, are often more valuable than less stable ones.
>
> To improve value propagation, we propose an algorithm that better captures value progression along a path. Given a path $\mathbf{P} = \{p_1, p_2, \dots, p_n\}$ with $n$ nodes, where each $p_i$ represents the value at node $i$, the total value is calculated by summing the increments between consecutive nodes with a length penalty. The increment between nodes $p_i$ and $p_{i-1}$ is $\Delta_i = p_i - p_{i-1}$. Negative increments are clipped at $-0.1$ and downweighted by 0.5. The final path value $V_{\text{final}}$ is:
>
> $$
> V_{\text{final}} = \sum_{i=2}^{n} \Delta_i \quad \text{if } \Delta_i \geq 0
> $$
>
> $$
> V_{\text{final}} = \sum_{i=2}^{n} \left( 0.5 \times \max(\Delta_i, -0.1) \right) \quad \text{if } \Delta_i < 0
> $$
>
> $$
> V_{\text{final}} = V_{\text{final}} - \lambda \times n
> $$
>
>
>
> where $n$ is the number of nodes in the path and $\lambda = 0.1$ is the penalty factor to discourage long paths. Through the ablation study in Section 5.5, we observe that our optimized backpropagation algorithm significantly improves the performance of MCTS reasoning, which is one of our contributions.
>
> As mentioned in Section 4.2, the node selection strategy UCT is a critical component. The UCT value is defined as:
>
> $
> UCT_j = \bar{X}_j + C \sqrt{\frac{\ln N}{N_j}}
> $
>
> where $\bar{X}_j$ is the average reward of taking action $j$, $N$ is the number of times the parent node has been visited, $N_j$ is the number of times node $j$ has been visited for simulation, and $C$ is a constant to balance exploitation and exploration. We found that in previous work, the exploration term ($C \sqrt{\frac{\ln N}{N_j}}$) often failed due to the parameter $C$ being inadequately tuned. For instance, prior work such as RAP-MCTS often assumed $C=1$ as a prior value, leading to almost no exploration. Please see Section 5.4, where the performance of RAP-MCTS is nearly identical to that of the Negative Control ($C=0$). Our improvement to this parameter is another one of our contributions.

---

> ### Author Response · Authors · 2024-11-21
>
> **Response(3/3)**
>
> The introduction of speculative decoding is merely a minor contribution. We have updated the title of the speculative decoding section to "Speculative Decoding as **'Free Lunch'**." As the title suggests, the acceleration benefits of speculative decoding are a “free lunch” provided by our novel reward model, which is designed based on the idea of contrastive decoding.
>
> Beyond these, our more significant contributions are:
>
> 1. As shown in the ablation study in Section 5.5, the performance of MCTS reasoning is almost entirely dependent on the reward model. Thus, our key contribution is designing a **novel and high-performance reward model** based on the idea of contrastive decoding.
>
> 2. We identified flaws in previous algorithms that combined multiple reward models and proposed a **linear combination algorithm based on prior statistical methods**, supplemented by an **online incremental update algorithm for mean and variance** to prevent distributional shifts.
>
> 3. In Section 5.6, we demonstrated that our reward model is **highly interpretable** compared to prior work by analyzing numerical distributions, quantile-reward mappings, Spearman correlation, Pearson correlation, and p-values. Consequently, **our SC-MCTS\*** is also highly interpretable since its performance is almost entirely determined by the reward model.
>
>
> **For question 2:**
>
> A* is a heuristic search algorithm primarily used for finding the shortest path from a starting point to a target point, while MCTS is a decision search algorithm based on stochastic simulations, commonly used in game decision-making or problems with large search spaces. These two algorithms employ entirely different heuristic approaches.
>
> The A* algorithm is suitable for pathfinding and graph search problems, such as robotic navigation and map routing. It requires clearly defined starting and target points, and its heuristic function must be designed with sufficient accuracy. In contrast, the MCTS algorithm is used for games, decision planning, reasoning, and other scenarios that require simulating complex dynamic systems. Unlike A*, MCTS does not require an explicitly defined target state and instead evaluates simulations to derive an optimal strategy.
>
> MCTS and A* are two fundamentally different heuristic algorithms designed for entirely different scenarios. In our problem settings, it is not possible to evaluate the performance of A*, whereas MCTS is highly suitable for multi-step reasoning with LLMs.
>
> **For question 3:**
>
> Our objective is not to study test-time scaling laws, as this diverges from the goal of our paper—to address significant gaps in prior work by analyzing and optimizing MCTS-based multi-step reasoning algorithms for LLMs, thereby improving their performance and interpretability.
>
> Our goal is to:
> 1. design novel and high-performance reward models and maximize the effectiveness of reward model combinations,
> 2. analyze and optimize the performance of various MCTS components, and
> 3. enhance the interpretability of MCTS reasoning.
>
> Research related to test-time scaling laws may be left for future work by others.
>
> **For question 4:**
>
> The superior performance of our algorithm is not simply due to using two models (expert and amateur models). Instead, it stems from our novel and high-performance reward model, designed based on the idea of contrastive decoding, which also demonstrates significantly better interpretability compared to reward models in prior work as demonstrated in Section 5.6, and this reward model requires the use of two models. Please refer to the ablation study in Section 5.5, where our reward model $R_{JSD}$, designed based on this concept, outperforms other reward models (e.g., log-likelihood and self-evaluation) by a significant margin.
>
> The improved reasoning accuracy of our SC-MCTS* algorithm is attributed to: (i) the introduction of this novel and high-performance reward model, based on the idea of contrastive decoding, and  (ii) identifying flaws in previous algorithms that combined multiple reward models and proposing a **linear combination algorithm based on prior statistical methods**, which maximizes the performance of multiple reward models.
>
> We are truly grateful that you took the time to thoroughly review our work! If you are satisfied with our response, we would greatly appreciate it if you could consider raising our score.

---

> ### Comment · Area_Chair_3ukZ · 2024-11-24
> **From AC.**
>
> Reviewer CGgt: if possible, can you reply to the rebuttal?

---

> ### Author Response · Authors · 2024-11-29
>
> Given that the deadline for rebuttal is fast approaching, please let us know if there are any remaining questions we can answer to address your concerns. Thank you!

---

> ### Comment · Reviewer_CGgt · 2024-11-30
>
> Thank you for the author's response. I will maintain my score.

---

> > ### Author Response · Authors · 2024-12-03
> >
> > Given that the deadline for rebuttal is approaching, please let us know if there are any remaining questions we can answer to address your concerns. We are truly grateful that you took the time to thoroughly review our work! If you are satisfied with our response, we would greatly appreciate it if you could consider raising our score.

---

> ### Author Response · Authors · 2024-11-30
>
> Thank you for your response. Could you point out where our rebuttal failed to address your concerns? We would greatly appreciate your suggestions.

---

### Official Review · Reviewer_2cCc · 2024-11-03

**Soundness:** 2
**Presentation:** 2
**Contribution:** 2
**Rating:** 3
**Confidence:** 3

**Summary:**

This  paper introduces Speculative Contrastive Monte Carlo Tree Search (SC-MCTS*), a novel reasoning algorithm designed to enhance the performance of Large Language Models (LLMs) in multi-step reasoning tasks. The authors aim to address the limitations of previous Monte Carlo Tree Search (MCTS) implementations, such as slower reasoning speed and insufficient analysis of core components, particularly the reward model. SC-MCTS* incorporates a new reward model grounded in contrastive decoding principles, speculative decoding for speed optimization, and refined node selection and back-propagation strategies.

**Strengths:**

The introduction of a reward model based on contrastive decoding, which emphasizes action-level evaluation, enhances interpretability and robustness.

**Weaknesses:**

* The impact of the evaluation of intermediate nodes on MCTS performance is significant but not discussed in depth. This oversight may lead to an incomplete understanding of the method's effectiveness.
* The novelty of applying MCTS to planning is somewhat diminished by the fact that this approach is already well-established in the literature. The paper would benefit from a more thorough comparison with existing methodologies to highlight its contributions.

**Questions:**

1. Could you clarify how planning and reasoning are defined in your framework? What specific characteristics differentiate them?
2. How does separating planning and reasoning lead to improved outcomes in your experiments?
3. The performance of MCTS appears to be influenced by how intermediate nodes are evaluated. Could you provide insights or analysis on this aspect within your methodology?
4. Are there specific scenarios or use cases where your proposed separation provides a distinct advantage over existing integrated approaches?

---

> ### Author Response · Authors · 2024-11-21
>
> **Response(1/2)**
>
> We would like to express our sincere gratitude for your careful reading and insightful feedback on our paper.
>
> **For weakness 1 and question 3,4:**
>
> We are uncertain about how the intermediate nodes you mentioned impact the performance of MCTS. Could you please reference relevant articles? In our Blocksworld multi-step reasoning dataset, we have exactly four actions and only one correct reasoning path, where each step holds the same level of importance. Therefore, in our problem settings, the performance impact of the intermediate nodes you described might not exist. Please see the details about the Blocksworld multi-step reasoning dataset in Appendix F, and refer to Appendix G for some examples of MCTS.
>
> **For weakness 2:**
>
> Thank you for your suggestions. As you mentioned, our contribution is not about applying MCTS to multi-step reasoning with LLMs. As noted in our paper, prior work has primarily used MCTS as a tool in some downstream reasoning tasks, with several technical reports published. However, quantitative studies and ablation analyses of MCTS components remain limited. Our contributions are as follows:
>
> 1. As shown in the ablation study in Section 5.5, the performance of MCTS reasoning is almost entirely dependent on the reward model. Thus, our key contribution is designing a **novel and high-performance reward model** based on the idea of contrastive decoding.
>
> 2. We identified flaws in previous algorithms that combined multiple reward models and proposed a **linear combination algorithm based on prior statistical methods**, supplemented by an **online incremental update algorithm for mean and variance** to prevent distributional shifts.
>
> 3. We observed that the UCT strategy may fail in prior work, potentially leading MCTS into dead ends. We improved this aspect.
>
> 4. We optimized the MCTS backpropagation algorithm to favor **steadily improving paths**, significantly enhancing performance.
>
> 5. We introduced **speculative decoding** as a “free lunch,” achieving an average of **52% speedup** in reasoning.
>
> 6. In Section 5.6, we demonstrated that our reward model is **highly interpretable** compared to prior work by analyzing numerical distributions, quantile-reward mappings, Spearman correlation, Pearson correlation, and p-values. Consequently, our SC-MCTS\* is also highly interpretable since its performance is almost entirely determined by the reward model.
>
> All these contributions are emphasized in the **Introduction Section**.

---

> ### Author Response · Authors · 2024-11-21
>
> **Response(2/2)**
>
> **For question 1,2:**
>
> In our framework, the LLM reasons via planning, enabling it to perform reasoning in a manner akin to human conscious planning. Specifically, the LLM reasons with principled planning (specifically Monte Carlo Tree Search), generating high-reward reasoning traces after effective exploration. During the reasoning process, the LLM is guided by our designed reward model, iteratively considering the most promising reasoning steps, anticipating future outcomes, and strategically constructing a reasoning tree. The estimated future rewards are then backpropagated to update the LLM’s beliefs about the current reasoning steps, guiding it to refine the reasoning by exploring better alternatives.
>
> Please refer to Appendix F for the setup and prompts of the Blocksworld multi-step reasoning dataset, where reasoning via planning is inseparable in our framework. As described in RAP-MCTS[1], in Blocksworld, states are defined as configurations of blocks, and actions are defined as behaviors such as moving a block, e.g., "pick up a block," described in natural language. Thus, the reasoning process can be described as a Markov Decision Process (MDP) by defining states and actions. Given the current state $s_t$, where $t=0, 1, \dots, T$, such as the initial state $s_0$, the LLM (acting as a reasoning agent) generates an action space by sampling from its generative distribution, $a_t \sim p(a | s_t, c)$, where $c$ represents an appropriate prompt (e.g., in-context demonstrations). Once an action is selected, the model predicts the next state $s_{t+1}$ during the reasoning process. Specifically, the LLM is repurposed to obtain a state transition distribution, $p(s_{t+1} | s_t, a_t, c')$, where $c'$ is another prompt designed to guide the LLM in generating the next state. For instance, in the setup of Blocksworld, the LLM generates a textual description $s_{t+1}$ of the new configuration of blocks based on the previous state $s_t$ and the action $a_t$.
>
> The continuation of this process generates a reasoning trace, represented as a sequence of interleaved states and actions $(s_0, a_0, s_1, \dots, a_{T-1}, s_T)$. The reasoning via planning process enables the LLM to achieve more grounded and coherent reasoning. It is important to note that the full reasoning trace is simulated entirely by the LLM itself, acting as a reasoning agent. This process is analogous to humans contemplating a potential plan in their minds. By introducing the ability to simulate future states using the world model, we can integrate principled planning algorithms to efficiently navigate the vast reasoning space.
>
> We are truly grateful that you took the time to thoroughly review our work! If you are satisfied with our response, we would greatly appreciate it if you could consider raising our score.
>
> [1]:Reasoning with Language Model is Planning with World Model

---

> > ### Comment · Reviewer_2cCc · 2024-12-01
> >
> > Thanks for detailed response. However, I still have some concerns that I'd like to address.
> >
> > As another reviewer mentioned some references, there are numerous related works using MCTS for LLMs, which have already discussed many of the problems mentioned in your paper. A more thorough comparison with existing methodologies would strengthen your contribution and help highlight the novelty of your approach.
> >
> > Regarding the interpretability of the contrastive reward model, I found Section 5.6 somewhat lacking in clarity. If we consider using the percentage of progress toward achieving the goal at a given step, the state-action value Q appears to be more interpretable than the proposed reward models. I believe a formal definition of interpretability, along with comparisons between more different reward models, is necessary to substantiate your claims.
> >
> > Minor point:
> > I also not fully disagree with the notion that reasoning is just a step in a MDP. Reasoning involves understanding and interpreting information to make informed judgments or solve problems. While planning may depend on reasoning, they are distinct cognitive processes. Clarifying how you define and differentiate planning and reasoning in your framework would be beneficial.

---

> > > ### Author Response · Authors · 2024-12-03
> > >
> > > Given that the deadline for rebuttal is approaching, please let us know if there are any remaining questions we can answer to address your concerns. We are truly grateful that you took the time to thoroughly review our work! If you are satisfied with our response, we would greatly appreciate it if you could consider raising our score.

---

> ### Comment · Area_Chair_3ukZ · 2024-11-24
> **From AC.**
>
> Reviewer 2cCc: if possible, can you reply to the rebuttal?

---

> ### Author Response · Authors · 2024-11-29
>
> Given that the deadline for rebuttal is fast approaching, please let us know if there are any remaining questions we can answer to address your concerns. Thank you!

---

> ### Author Response · Authors · 2024-12-01
>
> **Response(1/3)**
>
> Thanks for your valuable response!
>
> > As another reviewer mentioned some references, there are numerous related works using MCTS for LLMs, which have already discussed many of the problems mentioned in your paper. A more thorough comparison with existing methodologies would strengthen your contribution and help highlight the novelty of your approach.
>
> In our paper, we referenced numerous prior works on MCTS [1-7]. However, I may disagree with your statement that these works addressed many of the issues discussed in our paper. As mentioned in the Introduction Section, most of these prior works primarily used MCTS as a tool for downstream tasks, lacking quantitative analysis or ablation studies of all MCTS components. Our proposed SC-MCTS* focuses on the following issues:
>
> 1. One of our core contributions is the design of a high-performance reward model based on contrastive decoding. As shown in the ablation experiments in Section 5.5, its performance significantly surpasses the two reward models (log-likelihood and self-evaluation) of the baseline RAP-MCTS.
>
> 2. The baseline RAP-MCTS [4] we selected is widely recognized as a foundational work in the LLM MCTS domain. Recent related works have mainly built upon it for technical reports on other downstream tasks. No prior works identified the following issues:
>    - (i) The two reward models used by RAP-MCTS, log-likelihood and self-evaluation, exhibit very different distributions. For example, in the implementation of Llama-3.1-70B, the former's values are mainly distributed in the range (-580, -500), while the latter's values are mainly distributed in (-4, 0). RAP-MCTS combines these two reward models by directly adding them without any linear statistical methods or coefficients. This directly results in suboptimal combined effects. Our proposed Multi-RM method effectively addresses this issue through a prior-based statistical method and an online incremental update algorithm to prevent distribution drift. As demonstrated in the ablation experiments in Section 5.5, the Multi-RM method significantly improves performance. **There is no evidence that prior works discussed this issue.**
>    - (ii) The hyperparameter \(C\) of the UCT node selection strategy used in RAP-MCTS is an unbelievably default value of 1, as specified in the original UCT paper. Recall that the UCT value is defined as: $UCT_j= \bar{X}_j + C \sqrt{\frac{\ln N}{N_j}}$, where $\bar{X}_j$ is the average reward of taking action $j$, $N$ is the number of times the parent has been visited, and $N_j$ is the number of times node $j$ has been visited for simulation, $C$ is a constant to balance exploitation and exploration. $\bar{X}_j$, derived from the reward model, typically lies in the range (-600, 0). Setting $C$ to the default value of 1 almost entirely negates the exploration term. Figure 4 in Section 5.4 demonstrates this, showing that RAP-MCTS's performance with $C=1$ is nearly identical to that of the null control group ($C=0$). Our improved $C$ value, derived from a prior-based statistical method, significantly enhances performance. **There is no evidence that prior works discussed this issue.**
>
> 3. LLM MCTS is relatively slower than CoT because MCTS may require accessing hundreds of nodes, which equates to hundreds of rounds of dialogue, whereas CoT typically completes tasks in several rounds of dialogue. **There is no evidence that prior works discussed this issue.** By introducing a reward model based on contrastive decoding, we can leverage the speculative decoding acceleration of up to 52% as a "free lunch."

---

> ### Author Response · Authors · 2024-12-01
>
> **Response(2/3)**
>
> > Regarding the interpretability of the contrastive reward model, I found Section 5.6 somewhat lacking in clarity. If we consider using the percentage of progress toward achieving the goal at a given step, the state-action value Q appears to be more interpretable than the proposed reward models. I believe a formal definition of interpretability, along with comparisons between more different reward models, is necessary to substantiate your claims.
>
> Thank you for your valuable suggestion! To address your concern, we introduce the **Rank Information Coefficient (RIC) as a formal definition of interpretability** of the MCTS reward model. The IC measures the linear correlation between the reward value $R_a$ and the corresponding progress difference $\delta_a$. Information coefficient is defined as:
>
> $\text{Information Coefficient (IC)} = \frac{\sum_{i=1}^{n} (R_{a_i} - \overline{R_a})(\Delta_{a_i} - \overline{\Delta_a})}{\sqrt{\sum_{i=1}^{n} (R_{a_i} - \overline{R_a})^2 \sum_{i=1}^{n} (\Delta_{a_i} - \overline{\Delta_a})^2}}$
>
> Here, $R_{a_i}$ and $\delta_{a_i}$ are the reward value and progress difference for action $a_i$, respectively, and $\overline{R_a}$, $\overline{\delta_a}$ are their means.
>
> While IC effectively captures the linear relationship, it assumes that the reward values and progress differences follow a linear trend. However, in multi-step reasoning tasks, the relationship between $R_a$ and $\delta_a$ may not be strictly linear due to the complex nature of reasoning processes. This motivates us to use a **rank-based information coefficient (RIC)**, which measures the monotonic relationship instead of the linear correlation. The RIC is defined as:
>
> $$
> \text{Rank Information Coefficient (RIC)} = \frac{\sum_{i=1}^{n} (\text{Rank}(R_{a_i}) - \overline{\text{Rank}(R_a)})(\text{Rank}(\Delta_{a_i}) - \overline{\text{Rank}(\Delta_a)})}{\sqrt{\sum_{i=1}^{n} (\text{Rank}(R_{a_i}) - \overline{\text{Rank}(R_a)})^2 \sum_{i=1}^{n} (\text{Rank}(\Delta_{a_i}) - \overline{\text{Rank}(\Delta_a)})^2}}
> $$
>
> Here, $\text{Rank}(R_{a_i})$ and $\text{Rank}(\delta_{a_i})$ represent the ranks of $R_{a_i}$ and $\delta_{a_i}$, respectively. By focusing on ranks rather than raw values, the RIC is more robust to outliers and non-linear relationships, making it a better metric for evaluating interpretability in scenarios with non-linear or complex dynamics.
>
> ### Why RIC is Superior to IC
> 1. **Non-Linear Robustness**: RIC captures monotonic relationships, making it suitable for scenarios where the relationship between $R_a$ and $\delta_a$ is not strictly linear.
> 2. **Outlier Resistance**: By operating on ranks, RIC reduces the influence of extreme values in $R_a$ and $\delta_a$, ensuring more stable interpretability assessments.
> 3. **Action Prioritization**: In MCTS, the rank of an action’s reward value is often more critical for guiding search than the absolute reward value, aligning RIC closely with the algorithm’s decision-making process.
>
> We compute both IC and RIC for SC-MCTS* and compare them against other reward models. The experimental results are presented in table below:
>
> | **Reward Model**                                 | **RIC**  | **IC**   |
> |--------------------------------------------------|----------|----------|
> | $SC\text{-}MCTS^*$(ours)                               | 0.3559   | 0.3720   |
> | $R_{\text{JSD}}$                                 | 0.2942   | 0.3125   |
> | $R_{\text{LL}}$                                  | 0.1165   | 0.1206   |
> | $R_{\text{SE}}$                                  | 0.0745   | 0.0602   |
> | $RAP\text{-}MCTS_{(R_{LL} + R_{SE})}$            | 0.1225   | 0.1280   |
>
> From the table we can observe that SC-MCTS* achieves significantly higher RIC compared to other models, indicating stronger monotonic alignment between the reward values and progress differences. This substantiates our claim that SC-MCTS* provides highly interpretable reward signals, effectively guiding the reasoning process.
>
> The rank information coefficient serves as a formal metric, offering a rigorous and quantitative basis for formally evaluating the interpretability of reward models in MCTS reasoning. This addition strengthens the foundation of our claims and effectively addresses your concern.

---

> ### Author Response · Authors · 2024-12-01
>
> **Response(3/3)**
>
> > Minor point: I also not fully disagree with the notion that reasoning is just a step in a MDP. Reasoning involves understanding and interpreting information to make informed judgments or solve problems. While planning may depend on reasoning, they are distinct cognitive processes. Clarifying how you define and differentiate planning and reasoning in your framework would be beneficial.
>
> We sincerely appreciate your feedback and your nuanced perspective on the distinction between planning and reasoning.
>
> In our framework, we recognize reasoning as a broader cognitive process encompassing the interpretation of information, logical deductions, and problem-solving. Planning, on the other hand, is a specific subset of reasoning focused on formulating and sequencing actions to achieve a goal. Planning operationalizes reasoning by utilizing structured methodologies—such as MDP—to simulate, evaluate, and execute actions in a goal-directed manner.
>
> To clarify the differentiation within our framework:
>
> **Reasoning:** This involves understanding the problem context, interpreting the states, and generating potential solutions. For example, in Blocksworld, reasoning involves deducing that moving a specific block will progress toward achieving the target configuration.
>
> **Planning:** This entails the selection, sequencing, and execution of specific actions to achieve the goal, guided by an optimization framework. Within our implementation, planning is realized through MCTS, where states and actions are iteratively explored, evaluated, and refined based on their estimated rewards.
>
> While the two are interrelated, our methodology treats planning as the mechanism through which reasoning becomes actionable. This alignment allows the framework to not only simulate reasoning paths but also optimize them systematically to achieve high-reward outcomes.
>
> To address the notion of reasoning as a "step in an MDP," we emphasize that the planning process is not independent but intrinsically linked to reasoning. By using MCTS, we embed reasoning within planning by iteratively refining reasoning steps (e.g., state-action transitions) through the exploration-exploitation tradeoff. Thus, in our approach, reasoning is simulated as part of the iterative planning process, where each reasoning trace corresponds to an MDP trajectory.
>
> We would like to emphasize that several of our contributions have not been discussed in prior work, and our experiments have demonstrated their significant value for LLM MCTS reasoning. Once again, our contributions are as follows:
>
> 1. As shown in the ablation study in Section 5.5, the performance of MCTS reasoning is almost entirely dependent on the reward model. Thus, our key contribution is designing a **novel and high-performance reward model** based on the idea of contrastive decoding.
>
> 2. We identified flaws in previous algorithms that combined multiple reward models and proposed a **linear combination algorithm based on prior statistical methods**, supplemented by an **online incremental update algorithm for mean and variance** to prevent distributional shifts.
>
> 3. We observed that the UCT strategy may fail in prior work, **potentially leading MCTS into dead ends**. We improved this aspect with our prior statistical methods.
>
> 4. We optimized the MCTS backpropagation algorithm to favor **steadily improving paths**, significantly enhancing performance.
>
> 5. We demonstrated that our reward model is **highly interpretable** compared to prior work by analyzing numerical distributions, quantile-reward mappings, Spearman correlation, Pearson correlation, p-values, as well as the newly updated information coefficients and rank information coefficients. Consequently, our SC-MCTS* is also highly interpretable since its performance is almost entirely determined by the reward model.
>
> All the above updates will be included in the final version. We are truly grateful that you took the time to thoroughly review our work and provided such valuable suggestions! If you are satisfied with our response, we would greatly appreciate it if you could consider raising our score.
>
> [1] Accessing GPT-4 level Mathematical Olympiad Solutions via Monte Carlo Tree Self-refine with LLaMa-3 8B, https://arxiv.org/abs/2406.07394
>
> [2] Monte Carlo Tree Search Boosts Reasoning via Iterative Preference Learning, https://arxiv.org/abs/2405.00451
>
> [3] ReST-MCTS*: LLM Self-Training via Process Reward Guided Tree Search, https://arxiv.org/abs/2406.03816
>
> [4] Reasoning with Language Model is Planning with World Model, https://arxiv.org/abs/2305.14992
>
> [5] DeepSeek-Prover: Advancing Theorem Proving in LLMs through Large-Scale Synthetic Data, https://arxiv.org/abs/2405.14333
>
> [6] DeepSeek-Prover-V1.5: Harnessing Proof Assistant Feedback for Reinforcement Learning and Monte-Carlo Tree Search, https://arxiv.org/abs/2408.08152
>
> [7] Mutual Reasoning Makes Smaller LLMs Stronger Problem-Solvers, https://arxiv.org/abs/2408.06195

---

### Official Review · Reviewer_yZDg · 2024-11-08

**Soundness:** 2
**Presentation:** 2
**Contribution:** 3
**Rating:** 6
**Confidence:** 2

**Summary:**

This paper focuses on improving MCTS as a tool for reasoning in LLMs. The authors introduce a new method, which they call Speculative Contrastive MCTS, that refines previous methods. They create a composite reward model, introducing a method to properly normalize and combine three disparate reward signals. Further, they improve on previous work by modifying the exploration constant in UCT and better tuning the backpropogation method. Finally, they evaluate their model against Chain of Thought and RAP-MCTS (Hao et al 2023) on Blocksworld with various versions of Llama and GPT as the base model.

**Strengths:**

The authors introduce several improvements to previous MCTS methods, and verify these improvements via an ablation study. In particular, their method of combining three reward signals and adaptively weighting them is interesting and (to my knowledge) novel. They demonstrate a clear improvement on the Blocksworld dataset against RAP-MCTS as well as CoT.

**Weaknesses:**

**Methodology**
The authors only evaluate their method on the Blocksworld dataset. Showing results on other reasoning datasets such as GSM-8k, even if the experiment is limited in scope, would help show that the method generalizes to different types of tasks.

The authors could provide more detail as to how they chose hyperparameters, picked the reward clusters, etc.

In my opinion, the claim that the model is interpretable is not sufficiently motivated. In particular, I do not see how their observations on the distributions of the reward components make the reward more interpretable.

**Clarity**
There are significant grammatical errors which affect the clarity of the paper, and at some points the meaning of authors’ claims is ambiguous. Of course this is a minor issue as it can be easily fixed.

In the explanation of contrastive decoding, terms x_cont/x_pre and s_EXP/s_AMA are not defined.

**Questions:**

In the reward model section, it is stated: “Instead of using formal clustering algorithms like k-means, we manually define the regions based on the clear boundaries in the reward’s empirical distribution.” Can you provide more details on how these boundaries were found?

In section 5.4, how was the constant C found? Would this parameter need to be tuned for new reasoning tasks?

What makes the reward model more interpretable than previous methods? I see how the reward distribution can help evaluate the quality of the reward function, but I am not sure I understand how it can help interpret a particular set of reward values.

 It is worth noting that the paper has grammatical errors in the first sentences of the abstract and introduction, which should be fixed.
“We propose (S)peculative (C)ontrastive MCTS∗: a novel Monte Carlo Tree Search (MCTS) reasoning algorithm for Large Language Models (LLMs) [which] significantly improves both reasoning accuracy and speed”
“With the remarkable development of Large Language Models (LLMs), models such as o1 (OpenAI, 2024a) have now gained [a] strong ability [for] multi-step reasoning across complex tasks and [can] solve problems that are more difficult than previous scientific, code, and mathematical problems.”

---

> ### Author Response · Authors · 2024-11-20
>
> **Response(1/2)**
>
> We would like to express our sincere gratitude for your careful reading and insightful feedback on our paper.
>
> **For methodology weakness 1:**
>
> In the Future Work section, we have already explained why our experiments are conducted only on the Blocksworld multi-step reasoning dataset. This is because other reasoning datasets you mentioned, such as GSM8K and MATH, lack a unified step-segmentation mechanism, making it challenging to run MCTS reasoning in completion mode. All our experiments on Blocksworld were implemented using MCTS reasoning in completion mode, where each action can be easily segmented by a custom EOS token (e.g., for Llama-3, it is “\n[”). This allows us to construct search trees effortlessly, making the MCTS experiments highly controllable and better suited for studying our designed reward model at the action level. Blocksworld is an ideal dataset for studying LLM MCTS reasoning as it also provides a built-in ground truth verifier. So far, we have not identified other datasets with these properties.
>
> Adapting datasets like GSM8K for such experiments would require significant modifications. For example, defining the action space for mathematical reasoning tasks is extremely challenging. We might need to rewrite the entire dataset to enforce a unified step-segmentation template or fine-tune the model to enable natural and controllable step segmentation, which is left for future work. Thanks for your valuable feedback!
>
>
> **For methodology weakness 2:**
>
> We have added more relevant details to the Method Section and Parameters Section. Additionally, we have automated the process of picking the reward clusters and integrated it into the prior statistical data collection phase of Algorithm 1. Thanks for your valuable feedback!
>
>
> **For methodology weakness 3:**
>
> We have made significant updates to Section 5.6, Interpretability Study. We quantified the consistency between reward values and the proportion of positive $\Delta_a$ (a metric to quantify the contribution of each reasoning step towards the goal state, please see Section 5.6 for more details) using Spearman coefficients, Pearson coefficients, and p-values. As shown in the updated Figure 6, our reward model demonstrates significantly higher interpretability for $\Delta_a$ compared to the baseline. The mapping between reward value quantiles and the proportion of positive $\Delta_a$, as indicated by the color gradient from light to dark, is also much clearer. This strong alignment suggests that our reward model effectively captures progress toward the goal state, providing interpretable signals for action selection during MCTS reasoning. Thanks for your valuable feedback!
>
> **For clarity weakness:**
>
> We have revised and added several statements to make our claims clearer; please see lines 87–93. Additionally, we have provided more detailed explanations about the symbols you mentioned in our new version; please see lines 170–180. Thank you for your valuable feedback!
>
> **For question 1:**
>
> These boundaries are determined by the LLM checkpoint, and the boundaries of the reward model's value distribution may change when replacing the LLM. We have updated the reward model construction in Algorithm 1 by incorporating the definition of boundary regions into the prior statistical data calculation phase. This eliminates the need for manual definition. Thank you for your valuable feedback!
>
> **For question 2:**
>
> UCT value is defined as:
>
> $UCT_j= \bar{X}_j + C \sqrt{\frac{\ln N}{N_j}}$
>
> where $\bar{X}_j$ is the average reward of taking action $j$, $N$ is the number of times the parent has been visited, and $N_j$ is the number of times node $j$ has been visited for simulation, $C$ is a constant to balance exploitation and exploration.
>
> As discussed in Section 5.4, the constant $C$ is treated as a hyperparameter, tuned across the entire Blocksworld dataset. This parameter typically does not require adjustment for new reasoning tasks since we have already normalized $\bar{X}_j$. However, there might be cases where out-of-distribution values occur in new datasets. While it may be challenging to design an algorithm that ensures $C$ is always optimal, we have added the suggestion in Section 5.4: “After introducing new datasets, this hyperparameter may need to be re-tuned.” Additionally, we have updated our code to include the corresponding functionality. Thank you for your valuable feedback!

---

> ### Author Response · Authors · 2024-11-20
>
> **Response(2/2)**
>
> **For question 3:**
>
> We have made significant updates to Section 5.6, Interpretability Study. We quantified the consistency between reward values and the proportion of positive $\Delta_a$ (a metric to quantify the contribution of each reasoning step towards the goal state, please see Section 5.6 for more details) using Spearman coefficients, Pearson coefficients, and p-values. As shown in the updated Figure 6, our reward model demonstrates significantly higher interpretability for $\Delta_a$ compared to the baseline. The mapping between reward value quantiles and the proportion of positive $\Delta_a$, as indicated by the color gradient from light to dark, is also much clearer. This strong alignment suggests that our reward model effectively captures progress toward the goal state, providing interpretable signals for action selection during MCTS reasoning. Thanks for your valuable feedback!
>
> **For question 4:**
>
> Thanks for careful review and feedback! We have already fixed these grammatical errors in the new version of the paper.
>
>
> We have incorporated the all above updates into the new version. To be honest, your suggestions are extremely insightful and valuable. We are truly grateful that you took the time to thoroughly review our work! If you are satisfied with our response, we would greatly appreciate it if you could consider raising our score.

---

> ### Comment · Area_Chair_3ukZ · 2024-11-24
> **From AC.**
>
> Reviewer yZDg: if possible, can you reply to the rebuttal?

---

> ### Comment · Reviewer_yZDg · 2024-11-24
> **Comment**
>
> Thank you for the comprehensive response to my questions. I see that in the new version of the paper, you have made significant changes and addressed my concerns.
>
> I have also read through the discussions from the other reviewers. I will add that I appreciate the strength of this paper's contribution in making improvements to established methods. I will adjust my score.

---

### Official Review · Reviewer_9TgD · 2024-11-09

**Soundness:** 1
**Presentation:** 1
**Contribution:** 3
**Rating:** 3
**Confidence:** 3

**Summary:**

The paper proposes speculative contrastive MCTS algorithm. They redefined reward model (using an expert model) in the MCTS based on contrastive decoding. The paper also focus on improving different components of MCTS (e.g., node selection and back propagation) and achieves significant gain.

**Strengths:**

- I like the overall methodology of reconsidering different components of MCTS and improving them.
- Achieved empirical performance is impressive, especially compared to o1.

**Weaknesses:**

- The setup and requirements are not explained well. The introduction focuses on MCTS and its drawbacks but does not clearly explain the authors' goal, objective, and setup. Only in line 148 do they mention that the focus is on using existing LLMs to achieve better reasoning, but this is also not clearly presented. I would appreciate if the goal were explained in the introduction (along with a high-level setup and an example of a known expert). Additionally, in section 3.1, the authors could explain their assumption of access to an expert model.

- In Section 3.3, symbols are not explained. I had to go back and forth multiple times to speculate on the meanings of x_{count, pre}, s^{i}, V, {i}, etc.

- Throughout the paper, the authors kept mentioning a novel reward model, but it only becomes clear on page 5 that the objective is similar to distillation, which aims to mimic the expert. I would have considered adding a few papers focusing on distillation in the related works section. And explaining the setup before hand

- Since the LLM's are stochastic, can you also add uncetainity values across multiple runs in the experiments? That would be helpful.

**Questions:**

- Around line 241, what is n? and can you also write out JSD formula. Since the paper focus on action-level contrastive decoding and not the token-level decoding, I would expect to theoretically relate L_{CD} and Rewards (JSD, LL, SE) for these two (token level, action level) contrastive decoding. For e.g., with eqn in line 241 you can decompose p(x_i|x_{<i}) = \prod_{j=1}^{i} p(x_{j}|x_{<j}) into token level.

- line 97, perhaps rephrase with more details. "failed to function from our experiment" does not provide much details/intution

- line 84-90, perhaps rephrase it too. not clear "modes by clustering the prior distribution"

---

> ### Author Response · Authors · 2024-11-20
>
> We would like to express our sincere gratitude for your careful reading and insightful feedback on our paper.
>
> **For weakness 1:**
>
> We have added more detailed explanations of the goals, setup, and requirements at the end of the Introduction, please see lines 88–93. Our primary goal is to design novel and high-performance reward models for LLM MCTS reasoning and to maximize the performance potential of reward model combinations, as our ablation experiments in Section 5.5 demonstrate that MCTS performance is almost entirely determined by the reward models. Additionally, we have included related descriptions in the Method section (lines 204–207). Thanks for your valuable feedback!
>
> **For weakness 2:**
>
> Sorry for the confusion. We have added more detailed explanations about the symbols in our new version; please see lines 170–180. Thanks for your valuable feedback!
>
> **For weakness 3:**
>
> Our goal in designing the novel reward model is not similar to distillation but is instead inspired by the concept of contrastive decoding, it leverages the amateur model to enhance the reasoning ability of the expert model through our designed contrastive rewards. We have refined the relevant descriptions in the Method section to make this clearer. In our Related Work section, we have included several papers related to contrastive decoding; however, due to space constraints in the main text, we had to place them in the appendix. Additionally, we have mentioned at the end of the Related Work section: “For more detailed related work, please refer to Appendix B.” Thanks for your valuable feedback!
>
> **For weakness 4:**
>
> We have stated that we use greedy decoding in completion mode (lines 815–816), which means there is no uncertainty. Additionally, we have included the LLM hyperparameters in Appendix E.2 to emphasize this point.
>
> **For question 1:**
>
> Sorry for the confusion. We have added more detailed explanations about the symbols in our new version; please see lines 230–248. Thanks for your valuable feedback!
>
> **For question 2:**
>
> Please refer to Section 5.4, Figure 4, where we illustrate how the UCT strategy of the baseline (RAP-MCTS) fails to function. The accuracy of the baseline is nearly identical to that of the Negative Control (c=0). Additionally, we have added several practical examples of visualized search trees in Appendix G to more clearly demonstrate how the baseline fails to function and how our improved UCT strategy performs better. Thanks for your valuable feedback!
>
> **For question 3:**
>
> We have rephrased it; please refer to our revised paper. Thanks for your valuable feedback!
>
>
> We have incorporated the all above updates into the new version. To be honest, your suggestions are extremely insightful and valuable. We are truly grateful that you took the time to thoroughly review our work! If you are satisfied with our response, we would greatly appreciate it if you could consider raising our score.

---

> ### Comment · Area_Chair_3ukZ · 2024-11-24
> **From AC.**
>
> Reviewer 9TgD: if possible, can you reply to the rebuttal?

---

> ### Author Response · Authors · 2024-11-29
>
> Given that the deadline for rebuttal is fast approaching, please let us know if there are any remaining questions we can answer to address your concerns. Thank you!

---

> > ### Comment · Reviewer_9TgD · 2024-11-30
> >
> > Thanks for making edits for the paper to include more detail.
> > In my opinion W1 is not yet addressed and Q1 is only partially answered.
> > I would like to maintain my score.

---

> > > ### Author Response · Authors · 2024-12-03
> > >
> > > Given that the deadline for rebuttal is approaching, please let us know if there are any remaining questions we can answer to address your concerns. We are truly grateful that you took the time to thoroughly review our work! If you are satisfied with our response, we would greatly appreciate it if you could consider raising our score.

---

> ### Author Response · Authors · 2024-12-01
>
> Thank you for your response. Could you point out where our rebuttal of W1 and Q1 failed to address your concerns?
>
> > I would expect to theoretically relate $L_{CD}$ and Rewards (JSD, LL, SE) for these two (token level, action level) contrastive decoding. For example, with the equation in line 241, you can decompose $p(x_i|x_{<i}) = \prod_{j=1}^{i} p(x_{j}|x_{<j})$ into the token level.
>
> We are not entirely sure what is meant by theoretically relating $L_{CD}$. $L_{CD}$ represents the contrastive decoding objective and is not the target of our $R_{JSD}$. We introduced $L_{CD}$ to help readers better understand how it inspired our approach. Decomposing $p(x_i|x_{<i})$ into the token level is a method used in contrastive decoding. However, as described in Section 4.1, the reward model $R_{JSD}$ provides reward signals at the action level. Decomposing it into the token level might not have practical meaning. To avoid any potential misunderstanding, we will revised the description in Section 3.3.
>
> We would greatly appreciate your suggestions.

---

### Note · Authors · 2025-01-22

I have read and agree with the venue's withdrawal policy on behalf of myself and my co-authors.